

# Anisotropic scaling of the two-dimensional Ising model II: surfaces and boundary fields

**Hendrik Hobrecht and Alfred Hucht**⋆

Fakultät für Physik, Universität Duisburg-Essen, Lotharstr. 1, D-47048 Duisburg, Germany

⋆ fred@thp.uni-due.de

## Abstract

Based on the results published recently [SciPost Phys. 7, 026 (2019)], the influence of surfaces and boundary fields are calculated for the ferromagnetic anisotropic square lattice Ising model on finite lattices as well as in the finite-size scaling limit. Starting with the open cylinder, we independently apply boundary fields on both sides which can be either homogeneous or staggered, representing different combinations of boundary conditions. We confirm several predictions from scaling theory, conformal field theory and renormalisation group theory: we explicitly show that anisotropic couplings enter the scaling functions through a generalised aspect ratio, and demonstrate that open and staggered boundary conditions are asymptotically equal in the scaling regime. Furthermore, we examine the emergence of the surface tension due to one antiperiodic boundary in the system in the presence of symmetry breaking boundary fields, again for finite systems as well as in the scaling limit. Finally, we extend our results to the antiferromagnetic Ising model.



# 1  Introduction

In the first part of this work [1], denoted as I in the following, we calculated the free energy of the two-dimensional square lattice Ising model with periodic and antiperiodic boundary conditions (BCs) in both directions for anisotropic couplings $K^{\perp}$ and $K^{\parallel}$ in perpendicular and parallel direction, respectively (both in units of $k_{\mathrm{B}}T$ with Boltzmann constant $k_{\mathrm{B}}$). We introduced an associated scaling theory for the anisotropic case and calculated the corresponding finite-size scaling functions, as well as the surface tension induced by at least one antiperiodic BC. This second part is devoted to the influence of surfaces and boundary fields (BFs) and will pick up the thread of the surface tension again.

Twenty years after Onsager presented his famous solution of the infinitely large two-dimensional Ising model [2], the theory of scaling laws got into the focus of research, and with it the need to describe finite systems with surfaces, [3–7], possibly with applied surface fields [8–14]. On the other hand, phase interfaces, with an interface tension, arise in the Ising model when the order parameter is kept constant while the system transfers from the unordered to the ordered phase. These demixing transitions can be found in many systems, e.g., binary liquids like 2,6-lutidin/water mixtures or ternary mixtures like 3-methyl-pyridine/water/heavy water [15–18], which both have a so called *closed-loop* phase diagram, i.e., they have an upper and a lower critical point. Other examples are colloidal suspensions immersed in such binary liquids and cell membranes [19,20]. The binary and ternary systems are of special interest for the experimental measurement of the critical Casimir effect [21–23], especially for colloidal interactions, as the (lower) critical demixing point is near room temperature in contrast to liquid helium [24,25]. Additionally the surface preferences of the colloidal particles and the containers can be chemically tuned to be either hydrophilic or hydrophobic, which leads to a variety of experimental setups [26–32].

This rich behaviour can be explained theoretically as follows: Beneath its remarkable temperature sensitivity the Casimir force strongly depends on the BCs, and the possible combinations of boundary fields can be used to sub-classify the universal behaviour further [33,34]. For the Ising universality class, Dirichlet BCs are the most simple kind of surface one can apply. On the finite lattice they can be implemented in two ways: indirectly by using open boundaries, i.e., setting a line of couplings on the torus equal to zero, or more directly by applying a staggered BF to both ends of the cut, which leads to the doubly staggered BCs. The latter one is believed to be equivalent to the former one in the scaling limit [33] and was subject to several studies [35–38]. Note that a combination of open and staggered boundaries is known as Brascamp-Kunz BCs [39].

The universal finite-size scaling functions of the two-dimensional Ising model with various BCs are usually investigated in the two contrary limits of either thin films [40, 41], i.e., arbitrary temperature but restricted geometry, or due to conformal field theory (CFT) [42, 43], i.e., at the critical temperature but arbitrary aspect ratio of the cylinder, see [44] for a more detailed calculation. In fact the latter one can be used to calculate the scaling functions for arbitrary geometries due to the conformal invariance [45]. Nevertheless, the connection between those two cases, i.e., for arbitrary temperature *and arbitrary aspect ratio*, is not so well investigated in the literature, and this work ought to fill this gap. The only exceptions are the

results for the periodic torus [46] and the cylinder [47], while for open BCs in both directions this task was performed only recently [48, 49].

We first recall the main results of part I: First we generalised the Kasteleyn-Fisher mapping between the two-dimensional Ising model and the problem of closest-packed dimers to the case of arbitrary couplings. Enforcing translationally invariant BCs in one direction, the calculation of the partition function was reduced to a $2 \times 2$ transfer matrix method by two successive Schur reductions, making use of the dual couplings introduced by the self-duality of the two-dimensional Ising model on the square lattice without any BFs. To distinguish between periodic and antiperiodic BCs (denoted as (p) and (a), respectively) we introduced the parameters $\alpha$ and $\beta$ for the perpendicular and parallel direction, respectively. In principle both parameters can take arbitrary values on the interval $[-1, 1]$, where we focus on $\beta = \pm 1$ for (anti-)periodic BCs in parallel direction and $\alpha \in \{-1, 0, 1\}$ in perpendicular direction, where $\alpha = 0$ accounts for open boundaries. The general scaling theory for two-dimensional systems with at least one translationally invariant direction of section I.3 will be used exhaustingly in this paper, as well as the anisotropic scaling theory. To calculate the scaling limit of the corresponding free energy, i. e., the limit $L \to \infty$, $M \to \infty$ with $\rho \equiv L/M = const.$, the hyperbolic structure of the scaling form (I.4.21) of the Onsager dispersion (I.4.2) was used together with suitable counting polynomials (I.4.24) to calculate the sums in terms of complex contour integrals.

This paper will be structured as follows: First we introduce open BCs in perpendicular direction of the system, breaking its translational invariance. We will use this system to analyse the emergence of (a) surface contributions to the scaling functions, and (b) a surface tension due to antiperiodic boundaries in parallel direction. Afterwards we will introduce a BF at one of the cylinders surfaces, following the procedure of McCoy and Wu [50, chap. VI] for a homogeneous field. Afterwards we will use the same procedure to emulate a staggered BF and show that it has no contribution to the scaling limit, which is a first hint towards the equivalence of open and the staggered BCs. Subsequently we will return to the torus by coupling the BF to both surfaces of the cylinder to implement periodic $(++)$ and antiperiodic $(+-)$ symmetry-breaking BCs in perpendicular direction. Finally we will show the aforementioned equivalence between open and doubly staggered BCs. All our results are in perfect agreement with the conformal field theory (CFT) results [42, 45].

## 2 The open cylinder (oo)

For the cylindrical geometry with open BCs, denoted as (oo), we modify the Hamiltonian of (I.2.1) such that the reduced couplings are homogeneous, $K_{\ell,m}^{\delta} = K^{\delta} > 0$ with $\delta = \perp, \parallel$, and there is no coupling between the rows $m = 1$ and $m = M$,

$$\mathcal{H}^{(\text{oo},\text{p})} = -K^{\perp} \sum_{\ell=1}^{L-1} \sum_{m=1}^{M} \sigma_{\ell,m} \sigma_{\ell+1,m} - K^{\parallel} \sum_{\ell=1}^{L} \sum_{m=1}^{M} \sigma_{\ell,m} \sigma_{\ell,m+1}, \tag{2.1}$$

with Ising spin variables $\sigma_{\ell,m} \in \{-1, +1\}$ and (anti-)periodic BCs in parallel direction, $\sigma_{\ell,m+M} \equiv \pm \sigma_{\ell,m}$. Again we can rewrite the partition function using the high-temperature formulation to obtain a form suitable for the Pfaffian method, where the non-singular part now reads

$$Z_0^{(\text{oo},\text{p})} = \prod_{\ell=1}^{L-1} \prod_{m=1}^{M} \cosh K^{\perp} \prod_{\ell=1}^{L} \prod_{m=1}^{M} 2 \cosh K^{\parallel}. \tag{2.2}$$

For the singular part of the partition function we implement a procedure for our further calculations: We start with the $L \times L$ matrix $\tilde{\mathcal{C}}_L^{(\alpha=0)}$ from (I.2.33), for which it is easy to see

that, assuming $\beta \in \{+1, -1\}$, the open boundaries lead to a tridiagonal matrix, as $b_L \equiv 0$ due to the choice of $\alpha = 0$. On the other hand, if we want to apply a BF, we need an additional line of spins either left at $\ell = 0$ or right at $\ell = L + 1$, which is infinitely strong coupled. If the corresponding entry is at the right side of the system, we have $b_{L+1} = 0$ as we set $t_{L+1} \equiv 0$ in (I.2.33e). Thus all following matrices will be tridiagonal.

For a given tridiagonal matrix $\tilde{\mathcal{C}}_L^{(0)}$ the determinant can be computed with a recursion formula based on the Laplace expansion, so the determinant of a matrix of the form

$$\tilde{\mathcal{C}}_L^{(0)} = \begin{pmatrix} a_1 & b_1 & & & \\ b_1 & a_2 & b_2 & & \\ & b_2 & \ddots & \ddots & \\ & & \ddots & \ddots & b_{L-1} \\ & & & b_{L-1} & a_L \end{pmatrix} \tag{2.3}$$

may be calculated either by recursively following the diagonal upwards or downwards, where the latter one is more common. If we know the determinant of the associated submatrices, the recursion relations read

$$\det \tilde{\mathcal{C}}_\ell^{(0)} = a_{L+1-\ell} \det \bar{\mathcal{C}}_{\ell-1}^{(0)} - b_{L+1-\ell}^2 \det \bar{\mathcal{C}}_{\ell-2}^{(0)}, \tag{2.4a}$$

$$\det \tilde{\mathcal{C}}_\ell^{(0)} = a_\ell \det \tilde{\mathcal{C}}_{\ell-1}^{(0)} - b_{\ell-1}^2 \det \tilde{\mathcal{C}}_{\ell-2}^{(0)}, \tag{2.4b}$$

for the downward and the upward calculation, respectively, i.e., for $\tilde{\mathcal{C}}_{\ell-1}^{(0)}$ we delete the last row and column, while for $\bar{\mathcal{C}}_{\ell-1}^{(0)}$ we delete the first row and column instead. As we assume the BC in parallel direction to be (anti-)periodic, i.e., $\beta \in \{p, a\}$, the dependency on $\beta$ is fully covered by the eigenvalues $\varphi_m^{(\beta)}$ and their counterparts $\gamma_m^{(\beta)}$, see (I.4.2), so we will drop the explicit dependency on $\beta$ and write $\varphi_m \equiv \varphi_m^{(\beta)}$ and $\gamma_m \equiv \gamma_m^{(\beta)}$. As all boundary terms we will handle only appear in the entries $a_1$, $a_L$, and $b_{L-1}$, we may as well calculate the determinant of the submatrix representing the bulk without any surfaces (not even open ones) and then use the Laplace expansion to include the boundaries. The corresponding matrix $\mathcal{C}_L^{(0)}$, with factorised term $-b$, is a special case of (I.4.3) with $\alpha = 0$, and reads

$$\mathcal{C}_L^{(0)}(\varphi_m) = \begin{pmatrix} 2\cosh\gamma_m & -1 & & \\ -1 & \ddots & \ddots & \\ & \ddots & \ddots & -1 \\ & & -1 & 2\cosh\gamma_m \end{pmatrix}, \tag{2.5}$$

such that its determinant may be simply calculated with the transfer matrix approach (I.2.37) of section I.2. Therefore we diagonalise the transfer matrix

$$\mathcal{T}(\varphi_m) = \begin{pmatrix} 2\cosh\gamma_m & -1 \\ 1 & 0 \end{pmatrix} = X(\varphi_m)\,\mathbf{diag}(\mathrm{e}^{\gamma_m}, \mathrm{e}^{-\gamma_m})\,X^{-1}(\varphi_m), \tag{2.6}$$

with the unitary matrix

$$X(\varphi_m) = \begin{pmatrix} \mathrm{e}^{\gamma_m} & \mathrm{e}^{-\gamma_m} \\ 1 & 1 \end{pmatrix}, \tag{2.7}$$

and calculate the $L$-th power (cf. the dual expression (44) in [48])

$$\mathcal{T}^L(\varphi_m) = \frac{1}{\sinh\gamma_m} \begin{pmatrix} \sinh([L+1]\gamma_m) & -\sinh(L\gamma_m) \\ \sinh(L\gamma_m) & -\sinh([L-1]\gamma_m) \end{pmatrix} \tag{2.8}$$

to find

$$\det \mathcal{C}_L^{(0)}(\varphi_m) = \langle 1, 0 | \mathcal{T}^L | 1, 0 \rangle = \frac{\sinh([L+1]\gamma_m)}{\sinh \gamma_m}. \tag{2.9}$$

Note that (2.9) can also be related to (2.5) via Chebyshev polynomials of the second kind.

Now we can use (2.4) to separate the determinant into the bulk, the surface and the finite-size contributions, without any knowledge of the concrete boundaries, as

$$p^{(\alpha)}(\varphi_m) \det \mathcal{C}_{L-1}^{(0)}(\varphi_m) - q^{(\alpha)}(\varphi_m) \det \mathcal{C}_{L-2}^{(0)}(\varphi_m)$$
$$= e^{L\gamma_m} \frac{\eta_+^{(\alpha)}(\varphi_m)}{2 \sinh \gamma_m} \left[ 1 + \frac{\eta_-^{(\alpha)}(\varphi_m)}{\eta_+^{(\alpha)}(\varphi_m)} e^{-2L\gamma_m} \right], \tag{2.10}$$

with $p^{(\alpha)}(\varphi_m)$ and $q^{(\alpha)}(\varphi_m)$ being variables representing the boundary terms, and with

$$\eta_\pm^{(\alpha)}(\varphi_m) = \pm \left[ p^{(\alpha)}(\varphi_m) - q^{(\alpha)}(\varphi_m) e^{\mp \gamma_m} \right]. \tag{2.11}$$

We remark that the notation $x_\pm \equiv (x \pm x^{-1})/2$ from (I.2.34) does *not* apply for the quantity $\eta_\pm$. An extended calculation, which combines both (2.4a) and (2.4b), is necessary for the $(++)$ and $(+-)$ BCs, see section 3. It is easy to see that the bulk contribution stems solely from the term $e^{L\gamma_m}$, as it is the only term whose logarithm is explicitly linear in the length scale $L$ (plus – of course – possible prefactors of the matrix). Additionally we can factorise anything that is not exponentially decaying with $L$ to identify the surface contributions, namely

$$F_s^{(\alpha,\beta)}(L, M) = -\frac{1}{2} \sum_{\substack{0 \le m < 2M \\ m \, \text{even/odd}}} \ln \frac{\eta_+^{(\alpha)}(\varphi_m)}{2 \sinh \gamma_m}, \tag{2.12}$$

where the sum is over even/odd $m$ for $\beta = \mathsf{a}/\mathsf{p}$, leaving the remainder as residual finite-size contribution. Fortunately this last contribution then has the form

$$F_{\text{st,res}}^{(\alpha,\beta)}(L, M) = -\frac{1}{2} \sum_{\substack{0 \le m < 2M \\ m \, \text{even/odd}}} \ln \left[ 1 + \frac{\eta_-^{(\alpha)}(\varphi_m)}{\eta_+^{(\alpha)}(\varphi_m)} e^{-2L\gamma_m} \right], \tag{2.13}$$

which is similar to the form of the scaling functions of thin films [40]. Thus we only have to identify the boundary terms and insert them into (2.10) to obtain our desired results.

For open boundaries the Schur reduction in section I.2 follows the same steps as for the torus, but we choose $\alpha = 0$ or analogously $z_L = 0$, where both only appear as pair. As the cylindrical BCs forbid most kinds of transition cycles on the oriented lattice, only one Pfaffian is needed to yield the correct partition function. This also follows naturally from the toroidal geometry; as we choose $\alpha = 0$, two of the Pfaffians are equal as their only difference thus vanishes, additionally, the sign marked in Tab. I.1 between either the even or the odd part thus cancels the corresponding Pfaffians out, leading to only the odd case for periodic and the even case for antiperiodic BCs in parallel direction. From (I.2.33) we conclude that the resulting matrix reads

$$\mathcal{C}_L^{(\text{oo})}(\varphi_m) = \left( \begin{array}{c|c} \begin{matrix} a_1 & -1 \\ -1 & \end{matrix} & \begin{matrix} \\ \mathcal{C}_{L-1}^{(0)} \end{matrix} \end{array} \right), \tag{2.14a}$$

with the boundary term

$$a_1 = \frac{t_-}{z} \mu^+(\varphi_m), \tag{2.14b}$$

where $\mu^\pm(\varphi) = \cos\varphi \pm t_+/t_-$. We can use (2.4a) to calculate its determinant. The relevant terms then read

$$\eta_\pm^{(\text{oo})}(\varphi_m) \equiv \pm\left[\frac{t_-}{z}\mu^+(\varphi_m) - e^{\mp\gamma_m}\right] \tag{2.15a}$$

or equivalently by using the Onsager dispersion (I.4.2),

$$\eta_\pm^{(\text{oo})}(\varphi_m) = \pm\left[z t_- \mu^-(\varphi_m) + e^{\pm\gamma_m}\right]. \tag{2.15b}$$

Finally the determinant reads

$$\det \mathcal{C}_L^{(\text{oo})}(\varphi_m) = \frac{\eta_+^{(\text{oo})}(\varphi_m)e^{L\gamma_m} + \eta_-^{(\text{oo})}(\varphi_m)e^{-L\gamma_m}}{2\sinh\gamma_m}. \tag{2.16}$$

We conclude with a formula for the singular part of the partition function for $\beta \in \{\text{p}, \text{a}\}$ as

$$\frac{Z^{(\text{oo},\beta)}}{Z_0^{(\text{oo},\beta)}} = \frac{1}{2}\left(\frac{2z}{1+t_+}\right)^{\frac{LM}{2}} \prod_{\substack{0 \le m < 2M \\ m\,\text{even/odd}}} \det^{\frac{1}{2}}\mathcal{C}_L^{(\text{oo})}(\varphi_m), \tag{2.17}$$

where the odd product gives periodic and the even product antiperiodic BCs.

**Surface contribution and scaling functions**

First we will focus on the periodic case and, as done before for the torus, we may split up the (singular part of the) free energy

$$F^{(\text{oo},\text{p})}(L, M) = -\ln\frac{Z^{(\text{oo},\text{p})}}{Z_0^{(\text{oo},\text{p})}} \tag{2.18}$$

according to its geometric contributions, i. e., into the bulk contribution, which is exactly the same as in section I.4, the surface contribution for open boundaries, which we will focus on now, and the residual part,

$$F^{(\text{oo},\text{p})}(L, M) = F_{\text{b}}^{(\text{p})}(L, M) + F_s^{(\text{oo},\text{p})}(M) + F_{\text{st,res}}^{(\text{oo},\text{p})}(L, M). \tag{2.19}$$

The separation of the last section gives us immediately the surface free energy of the finite system

$$F_s^{(\text{oo},\text{p})}(M) = -\frac{1}{2}\sum_{\substack{0 < m < 2M \\ m\,\text{odd}}} \ln\frac{\eta_+^{(\text{oo})}(\varphi_m)}{2\sinh\gamma_m}, \tag{2.20}$$

and to calculate its thermodynamic limit

$$f_s^{(\text{oo})}(t, z) \equiv \lim_{M\to\infty} M^{-1}F_s^{(\text{oo},\text{p})}(M) \tag{2.21}$$

we apply the Euler-Maclaurin sum formula again, which yields

$$f_s^{(\text{oo})}(t, z) = -\frac{1}{4\pi}\int_0^{2\pi} d\varphi\,\ln\frac{\eta_+^{(\text{oo})}(\varphi)}{2\sinh\gamma(\varphi)}. \tag{2.22}$$

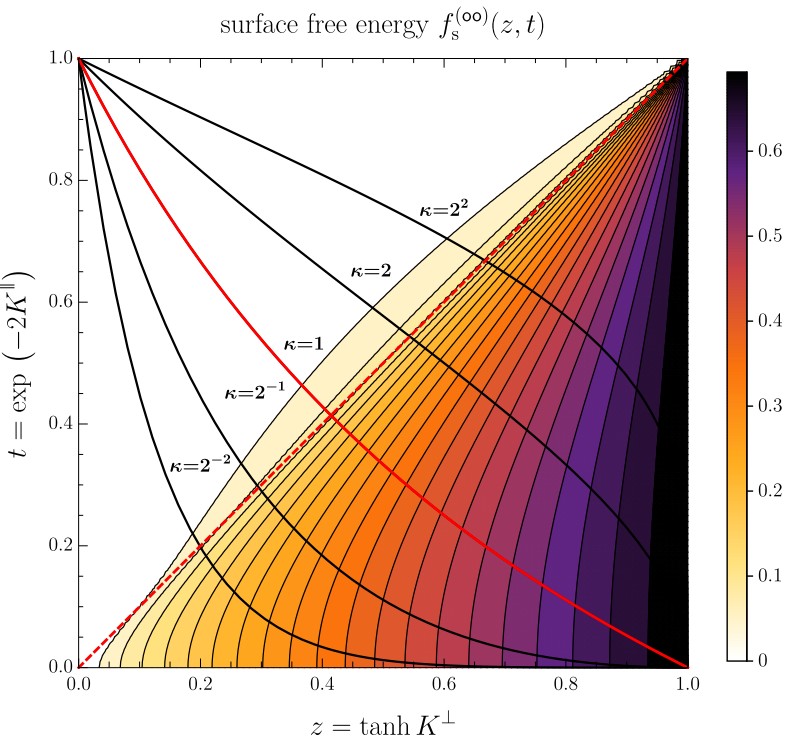

Figure 1: Bulk free energy density $f_s^{(\text{oo})}(z,t)$, see (2.22). The dashed line marks criticality for arbitrary anisotropy $\kappa = K^{\perp}/K^{\parallel}$. For fixed anisotropy the two couplings are connected by $t = (z^*)^{\kappa}$ and the black lines mark the run of the corresponding curve, where the isotropic case is shown in red.

Note that we were not able to solve this integral at the isotropic critical point, nevertheless its value is known exactly in terms of generalised zeta functions [49, (B.3)], and this form is numerical identical to both the exact value and the original expression by McCoy and Wu [50, (VI.4.24)], as well as the results by Baxter [51], with arbitrary precision. Fig. 1 shows the open surface free energy density for *both* open boundaries in contrast to the splitting with respect to each individual boundary in [50, chap. VI].

The scaling function of the residual surface contribution from open boundaries

$$F_{s,\text{res}}^{(\text{oo,p})}(M) \simeq \Theta_s^{(\text{oo,p})}(x_{\parallel}) \tag{2.23}$$

can be calculated in the same manner we already used to calculate the scaling function $\Theta_b(x_{\parallel})$. Therefore we need to make a series expansion of the associated term (see Appendix A for details) and find

$$\ln \frac{\eta_+^{(\text{oo})}(\varphi_m)}{2\sinh \gamma_m} \simeq \ln \frac{\Gamma_m + x_{\parallel}}{2\Gamma_m}, \tag{2.24}$$

with $\Gamma_m$ from (I.4.34). To calculate the summation of (2.24) we will use the hyperbolic parametrisation (I.4.25) again, but as there is an explicit dependency on $x_{\parallel}$ now in the formula we need to make a distinction between the ordered ($x_{\parallel} < 0$) and the unordered ($x_{\parallel} > 0$) phase.

In the unordered phase we use $x_{\parallel} = |x_{\parallel}|$ and replace the sum over $m$ by a contour integral as in (I.4.27) to find

$$\Theta_s^{(\text{oo,p})}(x_{\parallel} > 0) = \frac{1}{4\text{i}\pi} \oint_{\mathcal{C}>} \text{d}\omega \, |x_{\parallel}| \cosh \omega \ln \frac{1 + \text{sech}\,\omega}{2} \mathcal{K}_o^{\pm}(|x_{\parallel}|\sinh \omega), \tag{2.25}$$

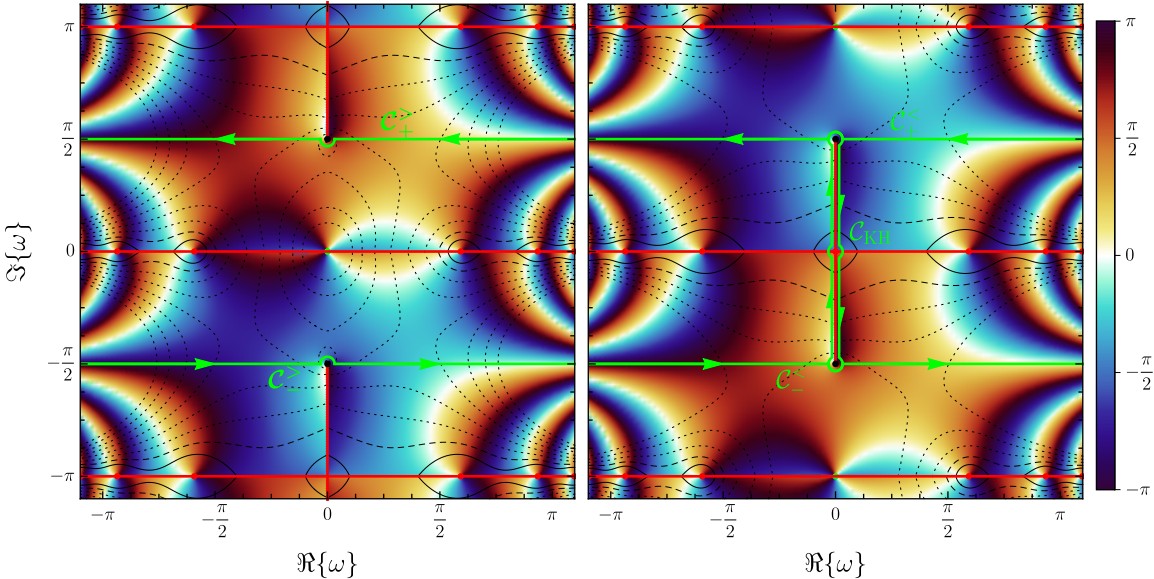

Figure 2: Complex structure of the contour integrands in the hyperbolic $\omega$-plane for the odd sum over $\ln\left[(\Gamma_m + x_\parallel)/(2\Gamma_m)\right]$ in the unordered phase (left, $x_\delta > 0$, (2.25)) and in the ordered phase (right, $x_\delta < 0$, (2.27)), together with the corresponding contours $\mathcal{C}^{\gtrless}$. The complex phase is colour coded from $-\pi$ to $\pi$, while the lines of constant absolute value $c$ are shown as black dotted ($c < 1$), dashed ($c = 1$) or solid ($c > 1$) lines, where $c$ are consecutive integer powers of two. Zeros (Poles) are marked as green (red) dots. Going from one to the other phase leads to a shift of one half-period of the functions along the imaginary axis, thus the contour $\mathcal{C}^< = \mathcal{C}^<_+ + \mathcal{C}^<_- + \mathcal{C}_{\mathrm{KH}}$ of the ordered phase is basically the reverse of the unordered phase $\mathcal{C}^> = \mathcal{C}^>_+ + \mathcal{C}^>_-$ plus the additional keyhole contour $\mathcal{C}_{\mathrm{KH}}$ around the logarithmic branch cut at $[i\pi/2, 3i\pi/2]$ (left) or at $[-i\pi/2, +i\pi/2]$ (right). Additionally the contours need to evade the branch points at $\omega = in\pi/2$ with $n \in \mathbb{Z}$. Note that there is a phase jump along the real axis and between every half-period due to the switching between the two counting kernels $\mathcal{K}^\pm_{\mathrm{o}}$, shown as red lines.

with the integration kernels (I.4.24); the integrand is shown on the left in Fig. 2. On the imaginary axis there is a $\pi$-periodic, logarithmic branch cut which will be crucial to the ordered phase for both the periodic and especially the antiperiodic case. For now we deform the contour with a semicircle around the two branch points and let their radius go to zero, which gives no contribution to the free energy. Thus we can calculate the contour integral again (see section I.4) by a simple shift in the integration variable by $\pm i\pi/2$, and find again that the upper and lower contour are identical with its real part being an odd function and its imaginary part being an even function, leaving the integral real. Again we can resubstitute to $\Phi$ and find, with $\Gamma$ from (I.4.19),

$$\Theta^{(\mathrm{oo,p})}_s(x_\parallel > 0) = -\frac{1}{4\pi} \int\limits_{-\infty}^{\infty} \mathrm{d}\Phi \, \frac{\Phi}{\Gamma} \arctan\frac{x_\parallel}{\Phi}\left[\tanh\frac{\Gamma}{2} - 1\right]. \tag{2.26}$$

For the ordered phase we use $x_\parallel = -|x_\parallel|$, which corresponds to shift of $i\pi$ in the $\omega$-plane, thus moving the logarithmic branch cut within our integration contour, see Fig. 2. Nevertheless, because of the $\pi$-periodicity of the hyperbolic parametrisation along the imaginary axes, we can reuse our result for the unordered phase, as we just change the direction of the corresponding parts of the contour, which together with the change to negative $x_\parallel$ leaves (2.26)

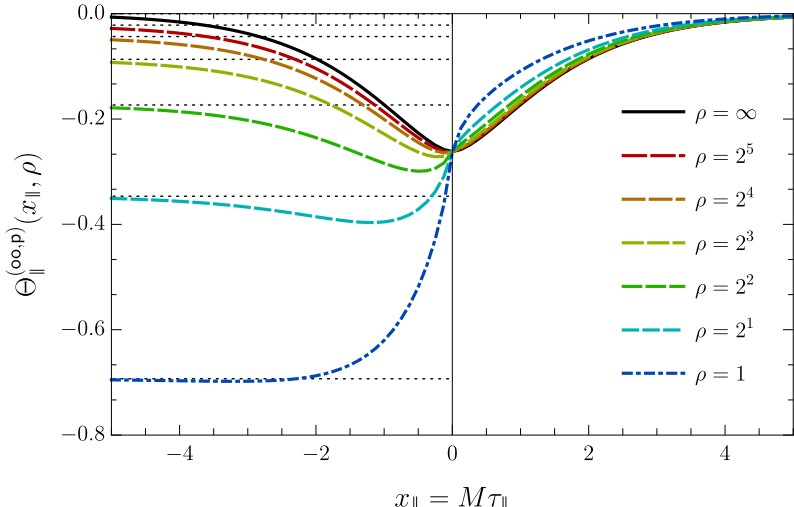

Figure 3: Scaling function $\Theta_\parallel^{(\mathrm{oo,p})}(x_\parallel, \rho)$ for different values of the aspect ratio $\rho \geq 1$. For larger $\rho$ the two open boundaries are getting farther away from each other, thus their interaction becomes irrelevant and the scaling function converges to the limiting case of $\Theta_\mathrm{b}^{(\mathrm{p})}(x_\parallel)$. For $\rho = 1$ the scaling function converges to $\ln 2$ for $x_\parallel \to -\infty$, which describes the difference between the two limiting procedures of the scaling limit and the thermodynamic limit.

unchanged. Thus we get an additional contribution from the keyhole integral around the branch cut, see Fig. 2,

$$\Theta_\mathrm{s}^{(\mathrm{oo,p})}(x_\parallel < 0) - \Theta_\mathrm{s}^{(\mathrm{oo,p})}(x_\parallel > 0)$$
$$= \frac{1}{4\mathrm{i}\pi} \oint_{\mathcal{C}_{\mathrm{KH}}} \mathrm{d}\omega \, |x_\parallel| \cosh \omega \ln \frac{1 - \operatorname{sech} \omega}{2} \mathcal{K}_\mathrm{o}^\pm(|x_\parallel| \sinh \omega). \quad (2.27)$$

Note that there is a jump in the phase along the real axis due to the switch between $\mathcal{K}_\mathrm{o}^+(\omega)$ and $\mathcal{K}_\mathrm{o}^-(\omega)$, which is beneficial for our calculation as the real part of the integrand is an odd function and its principal value with the logarithmic divergence at $\omega = 0$ does not contribute at all. The imaginary part of the integrand can be simplified by the principal value of the complex logarithm to

$$\Theta_\mathrm{s}^{(\mathrm{oo,p})}(x_\parallel < 0) - \Theta_\mathrm{s}^{(\mathrm{oo,p})}(x_\parallel > 0) = \int_0^{\mathrm{i}\frac{\pi}{2}} \mathrm{d}\omega \, |x_\parallel| \cosh \omega \, \mathcal{K}_\mathrm{o}^+(|x_\parallel| \sinh \omega) \quad (2.28\mathrm{a})$$

$$= \left[ \ln \mathcal{P}_\mathrm{o}^+(|x_\parallel| \sinh \omega) \right]_0^{\mathrm{i}\frac{\pi}{2}} \quad (2.28\mathrm{b})$$

$$= \ln \frac{1 + \mathrm{e}^{-|x_\parallel|}}{2}, \quad (2.28\mathrm{c})$$

where we first used that the the contributions from the four quadrants are all the same and then that the counting polynomials $\mathcal{K}_{\mathrm{e/o}}^\pm(\Phi)$ are defined as logarithmic derivative of the associated characteristic polynomial $\mathcal{P}_{\mathrm{e/o}}^\pm(\Phi)$, see (I.4.23) and (I.4.24). Again we see a contribution responsible for the effect due to the difference of the thermodynamic and the scaling limit caused by the broken symmetry of the system in the ordered phase. it gives a limiting value of $\ln 2$ for $x_\parallel \to -\infty$, which here solely stems from the keyhole integral around the logarithmic

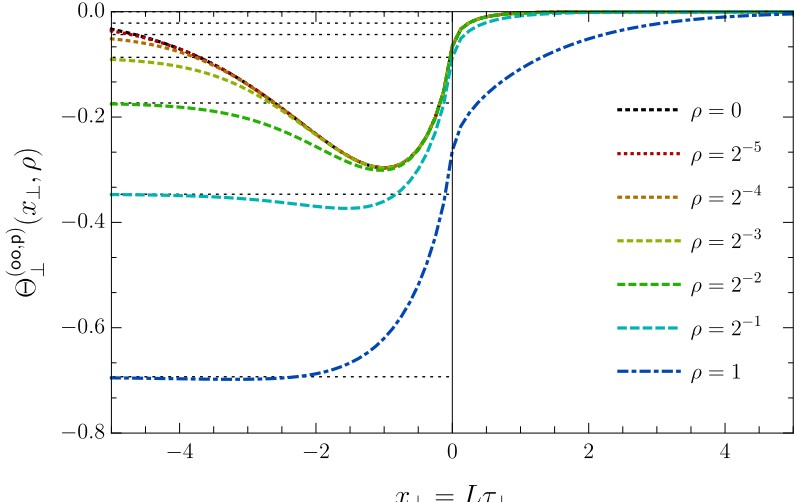

Figure 4: Scaling function $\Theta_\perp^{(oo,p)}(x_\perp,\rho)$ for different values of the aspect ratio $\rho \leq 1$. For $\rho \to 0$ it converges against the well-known case of the thin film $\Theta_\perp^{(oo)}(x_\perp)$, see (2.56), marked as dotted black line.

branch cut. Thus we find the scaling function of the residual contribution to the open BC free energy to be

$$\Theta_s^{(oo,p)}(x_\parallel) = -\frac{1}{4\pi}\int_{-\infty}^{\infty} d\Phi \, \frac{\Phi}{\Gamma} \arctan\frac{x_\parallel}{\Phi}\left[\tanh\frac{\Gamma}{2} - 1\right] + H(-x_\parallel)\ln\frac{1+e^{-|x_\parallel|}}{2}. \qquad (2.29)$$

For the finite-size contribution, we proceed likewise and find for the series expansion

$$\frac{\eta_-^{(oo)}(\varphi_m)}{\eta_+^{(oo)}(\varphi_m)} \simeq \frac{\Gamma_m - x_\parallel}{\Gamma_m + x_\parallel}, \qquad (2.30)$$

see again Appendix A, with which we conclude with a product analogous to (I.4.33)

$$P_{e/o}^{(oo)}(x_\parallel,\rho) = \prod_{\substack{m>0\\ m\,\text{even/odd}}}^{\infty}\left[1 + \frac{\Gamma_m - x_\parallel}{\Gamma_m + x_\parallel}e^{-2\rho\,\Gamma_m}\right] \qquad (2.31)$$

and a residual strip free energy scaling function

$$\Psi^{(oo,p)}(x_\parallel,\rho) = -\ln P_o^{(oo)}(x_\parallel,\rho), \qquad (2.32)$$

where we have already taken the square root by using only half of the eigenvalue spectrum.

Combining all these three contributions, the scaling function for the open cylinder with periodic BCs thus reads

$$\rho\,\Theta_\parallel^{(oo,p)}(x_\parallel,\rho) = \rho\,\Theta_b^{(p)}(x_\parallel) + \Theta_s^{(oo,p)}(x_\parallel) + \Psi^{(oo,p)}(x_\parallel,\rho) \qquad (2.33)$$

and is depicted in Fig. 3 for different values of $\rho$, as well as its counterpart for the perpendicular direction in Fig. 4.

**Domain wall**

The partition function of the cylindrical system with open boundaries in perpendicular direction and antiperiodic BCs in parallel direction differs from the one with periodic BCs only in the set over which the product is performed, namely the odd numbers for the latter and the even numbers for the former case. Thus we find

$$\frac{Z^{(\text{oo,a})}}{Z_0^{(\text{oo,a})}} = \frac{1}{2}\left(\frac{2z}{1+t_+}\right)^{\frac{LM}{2}} \prod_{\substack{0 \le m < 2M \\ m\,\text{even}}} \det^{\frac{1}{2}} \mathcal{C}_L^{(\text{oo})}(\varphi_m) \tag{2.34a}$$

$$= \frac{1}{2z^L}\left(\frac{2z}{1+t_+}\right)^{\frac{LM}{2}} \prod_{m=1}^{M/2-1} \det \mathcal{C}_L^{(\text{oo})}(\varphi_{2m}), \tag{2.34b}$$

where the additional factor $z^{-L}$ stems from the elimination of the square root: due to the symmetries of the product around $\varphi = 0$ and $\varphi = \pi$ every term but the ones for $\varphi = 0$ and $\varphi = \pi$ appears twice. Thus we have to calculate them separately and find

$$\det \mathcal{C}_L^{(\text{oo})}(0) = \left(\frac{t}{z}\right)^L, \tag{2.35a}$$

$$\det \mathcal{C}_L^{(\text{oo})}(\pi) = \left(\frac{1}{zt}\right)^L. \tag{2.35b}$$

In general the thermodynamic limits of the bulk and the surface contribution do not change despite the shifted summations. Thus the difference to the case of periodic BCs gives the additional energy due to the formation of a domain wall, and we can easily calculate the surface tension $\sigma^{(\text{oo,a})}(L, M)$ as

$$\sigma^{(\text{oo,a})}(L, M) = L \ln z - \sum_{m=1}^{M-1} (-1)^m \ln \det \mathcal{C}^{(\text{oo})}(\varphi_m) \tag{2.36}$$

and – just as usual – we can decompose it into the three parts according to bulk, surface, and finite-size contribution,

$$\sigma_{\text{b,res}}^{(\text{a})}(M) = \ln z - \sum_{m=1}^{M-1} (-1)^m \gamma_m, \tag{2.37a}$$

$$\sigma_{\text{s,res}}^{(\text{oo,a})}(M) = -\sum_{m=1}^{M-1} (-1)^m \ln \frac{\eta_+^{(\text{oo})}(\varphi_m)}{2 \sinh \gamma_m}, \tag{2.37b}$$

$$\sigma_{\text{st,res}}^{(\text{oo,a})}(L, M) = -\sum_{m=1}^{M-1} (-1)^m \ln \left[ 1 + \frac{\eta_-^{(\text{oo})}(\varphi_m)}{\eta_+^{(\text{oo})}(\varphi_m)} e^{-2L\gamma_m} \right], \tag{2.37c}$$

which each fulfils a scaling relation according to

$$L\,\sigma_{\text{b,res}}^{(\text{a})}(M) \simeq \rho\,\Sigma_{\text{b}}^{(\text{a})}(x_{\parallel}), \tag{2.38a}$$

$$\sigma_{\text{s,res}}^{(\text{oo,a})}(M) \simeq \Sigma_{\text{s}}^{(\text{oo,a})}(x_{\parallel}), \tag{2.38b}$$

$$\sigma_{\text{st,res}}^{(\text{oo,a})}(L, M) \simeq \Sigma_{\text{strip}}^{(\text{oo,a})}(x_{\parallel}, \rho), \tag{2.38c}$$

with the total surface tension scaling function

$$\rho\,\Sigma_{\parallel}^{(\text{oo,a})}(x_{\parallel}, \rho) = \rho\,\Sigma_{\text{b}}^{(\text{a})}(x_{\parallel}) + \Sigma_{\text{s}}^{(\text{oo,a})}(x_{\parallel}) + \Sigma_{\text{strip}}^{(\text{oo,a})}(x_{\parallel}, \rho). \tag{2.39}$$

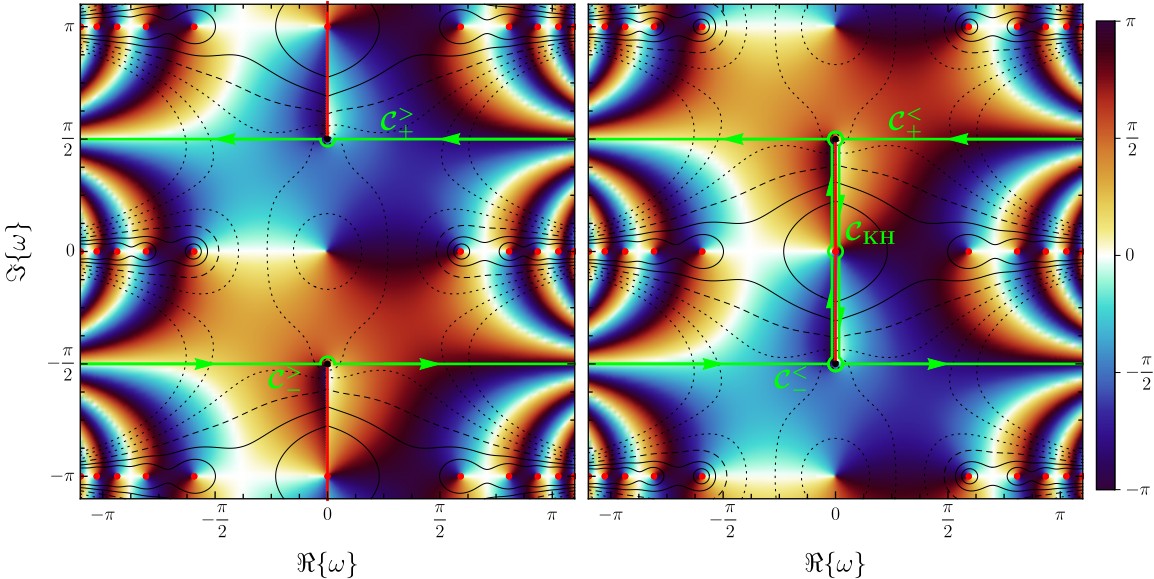

Figure 5: Complex structure of the contour integrands in the hyperbolic $\omega$-plane for the alternating sum over the $\ln\left[(\Gamma_m + x_\parallel)/(2\Gamma_m)\right]$ in the unordered phase (left, $x_\delta > 0$, (2.47a)) and in the ordered phase (right, $x_\delta < 0$, (2.49a)), together with the corresponding contours $\mathcal{C}^{\gtrless}$. In the unordered phase, the pole at $\omega = 0$ and the logarithmic cut at $[-\mathrm{i}\pi/2, +\mathrm{i}\pi/2]$ have to be excluded in order to fit with the related summation in (2.37b), thus the additional keyhole contour $\mathcal{C}_{\mathrm{KH}}$. For the colour coding see Fig. 2.

With the surface tension being the difference between the antiperiodic and the periodic case, the scaling functions of the further one are easily obtained as

$$\Theta_\parallel^{(\mathrm{oo,a})}(x_\parallel, \rho) = \Theta_\parallel^{(\mathrm{oo,p})}(x_\parallel, \rho) + \Sigma_\parallel^{(\mathrm{oo,a})}(x_\parallel, \rho), \tag{2.40}$$

and since the surface tension decomposes into its three parts just as the open cylinder, the scheme applies here, too.

To calculate the three contributions, we use the scaling limits in combination with the hyperbolic parametrisation. The counting polynomial is simply the difference of the even and the odd one $\delta\mathcal{K}$ of (I.4.24c). We start with the bulk contribution, as it contains the additional boundary terms from $\varphi = 0$ and $\varphi = \pi$, which contains the characteristic linear divergence of the scaling function in the ordered phase. Afterwards we will take a close look to the surface contribution, as the calculation in the hyperbolic parametrisation involves another keyhole integral corresponding to a logarithmic correction to the linear divergence. The strip free energy scaling function then is easy again, as its best converging form is again an infinite product.

For the bulk contribution to the surface tension scaling function, we first notice the great similarity to the finite form of $\delta\Theta_{\mathrm{b}}(x_\parallel)$ from the calculation for the torus in section I.4. Thus we rewrite the alternating sum as

$$-L \sum_{m=1}^{M-1} (-1)^m \gamma_m = \frac{L}{2}(\gamma_0 + \gamma_M) - \frac{L}{2} \sum_{m=0}^{2M-1} (-1)^m \gamma_m, \tag{2.41}$$

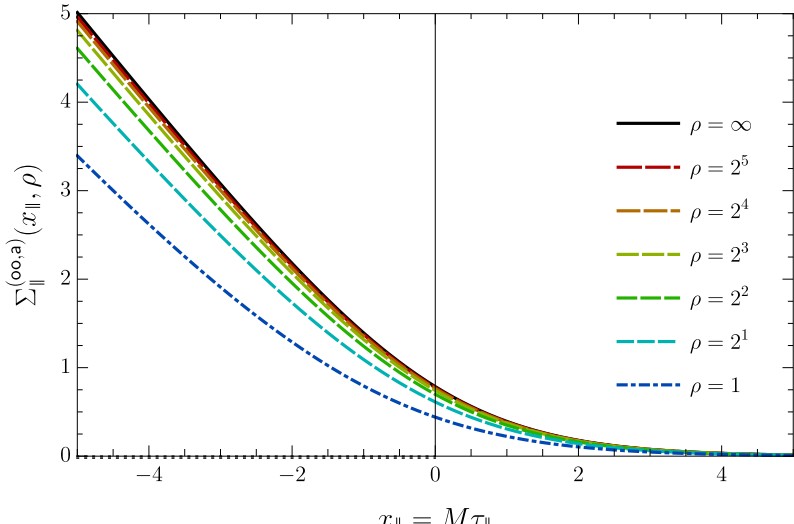

Figure 6: Scaling function $\Sigma_{\parallel}^{(\text{oo,a})}(x_{\parallel},\rho)$ of the surface tension for different values of the aspect ratio $\rho \geq 1$. For $\rho \rightarrow \infty$, it converges rather fast to its limiting case $\Sigma_{\text{b}}^{(\text{a})}(x_{\parallel})$ with the characteristic linear divergence in the ordered phase $x_{\parallel} < 0$.

where we used the symmetry of the $\gamma_m$. With the boundary values of $\gamma$,

$$\gamma_0 = -\ln(zt), \tag{2.42a}$$

$$\gamma_M = \text{sgn}(t-z)\ln\frac{t}{z}, \tag{2.42b}$$

we find the correction to be

$$L\ln z + \frac{L}{2}\left[\text{sgn}(t-z)\ln\frac{t}{z} - \ln(zt)\right] = -L\ln\frac{t}{z}H(z-t). \tag{2.43}$$

Using (I.4.17) we can go to the scaling limit by a series expansion around $M \rightarrow \infty$ to find

$$-L\ln\frac{t}{z}H(z-t) \simeq -\rho x_{\parallel}H(-x_{\parallel}), \tag{2.44}$$

with the Heaviside step function $H(x) \equiv \frac{\text{d}}{\text{d}x}\max\{0,x\}$, which is simply the linear diverging term we expected. Thus we can conclude for the bulk contribution with

$$\Sigma_{\text{b}}^{(\text{a})}(x_{\parallel}) = \delta\Theta_{\text{b}}(x_{\parallel}) - x_{\parallel}H(-x_{\parallel}), \tag{2.45}$$

and the bulk contribution for the antiperiodic case reads

$$\Theta_{\text{b}}^{(\text{a})}(x_{\parallel}) = \Theta_{\text{b}}^{(\text{p})}(x_{\parallel}) + \Sigma_{\text{b}}^{(\text{a})}(x_{\parallel}) \tag{2.46a}$$

$$= -\frac{1}{2\pi}\int_{-\infty}^{\infty}\text{d}\Phi\,\ln\left[1-\text{e}^{-\Gamma}\right] - x_{\parallel}H(-x_{\parallel}). \tag{2.46b}$$

Now we turn to the surface contribution and use (2.24) together with the alternating counting polynomial $\delta\mathcal{K}(\Phi)$ in the hyperbolic parametrisation, but here we need to exclude any pole at $\omega = 0$ as the sum only runs over the the positive numbers. Nevertheless, we can expand the contour to the negative half-plane, which simply counts every pole twice. Because of the alternating character of the integration kernel $\delta\mathcal{K}$ the integrals are convergent, see

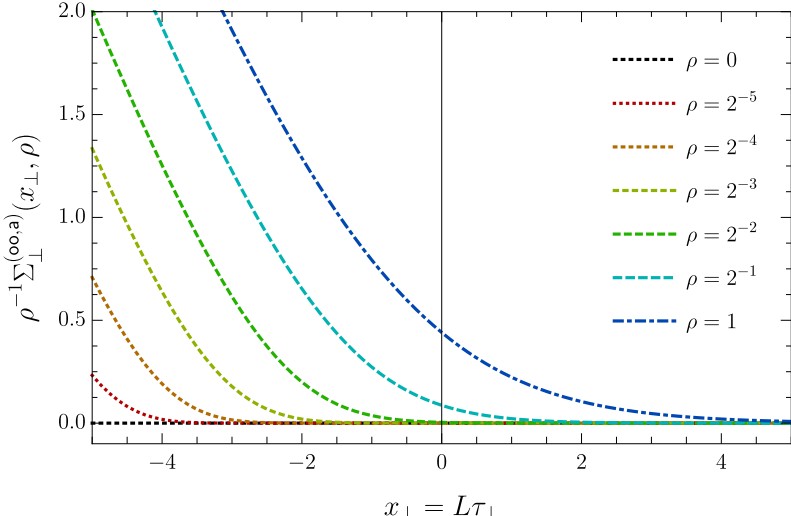

Figure 7: Scaling function $\Sigma_\perp^{(\text{oo},\text{a})}(x_\perp, \rho)$ for different values of the aspect ratio $\rho \leq 1$. With shrinking aspect ratio $\rho$ the influence of the domain wall vanishes and becomes zero in the limit of thin films.

Fig. 2, but, as for the open cylinder with periodic boundaries, we need to distinguish between the ordered and the unordered phase. For the unordered phase, there is no pole at $\omega = 0$, as the integrand has an associated zero, and thus, by the usual shift of the contour, we get

$$\delta\Theta_s^{(\text{oo})}(x_\parallel > 0) = \frac{1}{2i\pi} \oint_{\mathcal{C}_>} d\omega \, |x_\parallel| \cosh\omega \ln\frac{1 + \operatorname{sech}\omega}{2} \delta\mathcal{K}\left(|x_\parallel|\sinh\omega\right), \tag{2.47a}$$

$$= -\frac{1}{2\pi} \int_{-\infty}^{\infty} d\Phi \, \frac{\Phi}{\Gamma} \arctan\frac{x_\parallel}{\Phi} \operatorname{csch}\Gamma, \tag{2.47b}$$

where we substituted back to the $\Phi$-plane like done before. The corresponding complex structure is shown in Fig. 5.

In the ordered phase we have to face not only the logarithmic cut with the logarithmic divergence, but due to the integration kernel $\delta\mathcal{K}(\Phi) = \csc\Phi$ there is now an additional pole at the very same position. Additionally the other poles shift towards the origin for growing $|x_\parallel|$, forming a branch cut for $|x_\parallel| \to \infty$, and in order to exclude this crude construct from the desired integral we need to make a keyhole integral again. Fortunately the integral over the rest of the contour $\mathcal{C}_<$ is again the same as for the unordered phase because of the cancelation of the switching signs. To calculate the keyhole integral we use the series expansion of the corresponding integration kernel

$$\delta\mathcal{K}(\Phi) = \Phi^{-1} + 2\Phi \sum_{k=1}^{\infty} \frac{(-1)^k}{k^2\pi^2 - \Phi^2} \tag{2.48}$$

and separate the integrations with $\delta\mathcal{K}(\Phi) - \Phi^{-1}$ and $\Phi^{-1}$ as integration kernels along the

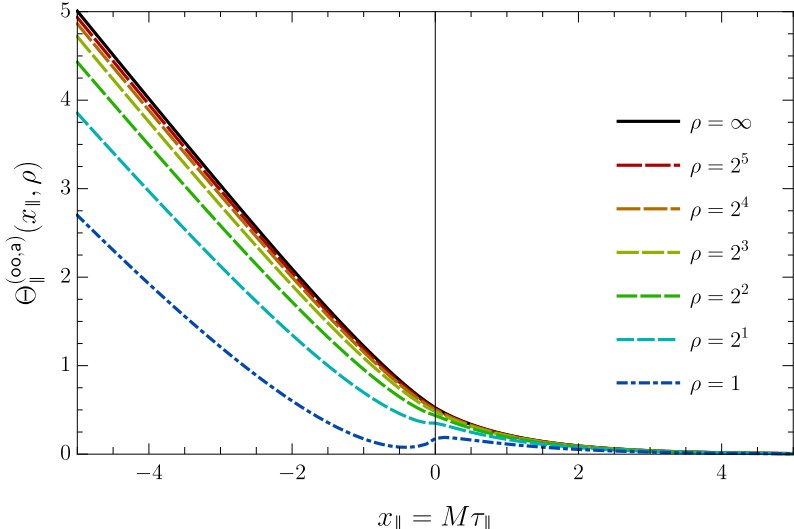

Figure 8: Scaling function $\Theta_{\parallel}^{(\text{oo},\text{a})}(x_{\parallel},\rho)$ of the free energy for different values of the aspect ratio $\rho \geq 1$. For $\rho \to \infty$, it converges rather fast to its limiting case of the dominant surface tension $\Sigma_{\text{b}}^{(\text{a})}(x_{\parallel})$, with the characteristic linear divergence in the ordered phase $x_{\parallel} < 0$.

keyhole contour. Without the pole, the logarithmic divergence again cancels out and we find

$$\frac{1}{4\text{i}\pi} \oint_{\mathcal{C}_{\text{KH}}} \text{d}\omega \, |x_{\parallel}| \cosh\omega \ln\frac{1-\text{sech}\,\omega}{2} \left[ \csc\left(|x_{\parallel}|\sinh\omega\right) - |x_{\parallel}|^{-1}\text{sech}\,\omega \right] \tag{2.49a}$$

$$= \int_{0}^{\text{i}\frac{\pi}{2}} \text{d}\omega \left[ |x_{\parallel}| \cosh\omega \, \delta\mathcal{K}\left(|x_{\parallel}|\sinh\omega\right) - \coth\omega \right] \tag{2.49b}$$

$$= \left[ \ln\mathcal{P}_{\text{e}}^{+}\left(|x_{\parallel}|\sinh\omega\right) - \ln\mathcal{P}_{\text{o}}^{+}\left(|x_{\parallel}|\sinh\omega\right) - \ln\sinh\omega \right]_{0}^{\text{i}\frac{\pi}{2}} \tag{2.49c}$$

$$= \ln\left[\frac{2}{x_{\parallel}} \tanh\frac{x_{\parallel}}{2}\right], \tag{2.49d}$$

where we again used the symmetry along the keyhole contour. A careful study of the remaining integral leaves us with

$$\frac{1}{2\text{i}\pi} \oint_{\mathcal{C}_{\text{KH}}} \text{d}\omega \, \coth\omega \ln\frac{1-\text{sech}\,\omega}{2} = -\ln 2, \tag{2.50}$$

which we can combine to

$$\delta\Theta_{\text{s}}^{(\text{oo})}(x_{\parallel}) = -\frac{1}{2\pi} \int_{-\infty}^{\infty} \text{d}\Phi \, \frac{\Phi}{\Gamma} \arctan\frac{x_{\parallel}}{\Phi} \text{csch}\,\Gamma + H(-x_{\parallel})\ln\left[\frac{1}{x_{\parallel}}\tanh\frac{x_{\parallel}}{2}\right]. \tag{2.51}$$

Hence we find for the surface tension simply

$$\Sigma_{\text{s}}^{(\text{oo},\text{a})}(x_{\parallel}) = \delta\Theta_{\text{s}}^{(\text{oo})}(x_{\parallel}), \tag{2.52}$$

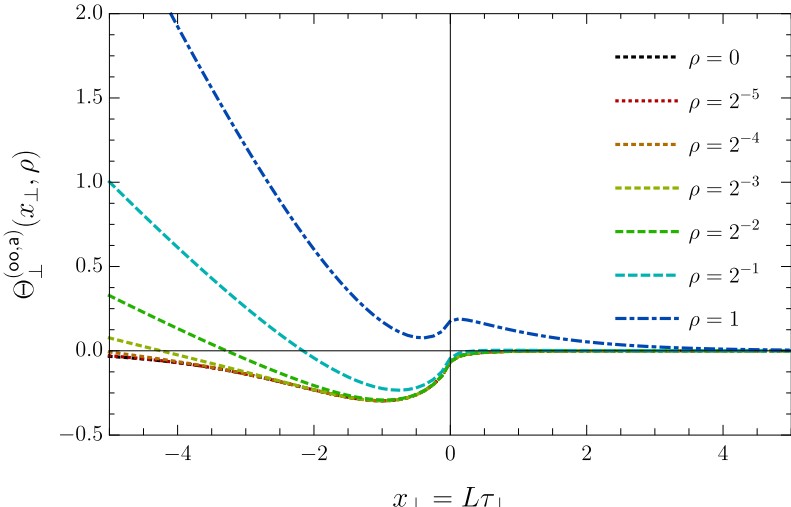

Figure 9: Scaling function $\Theta_\perp^{(\text{oo,a})}(x_\perp,\rho)$ for different values of the aspect ratio $\rho \leq 1$. With shrinking aspect ratio the influence of the domain wall vanishes and thus the function converges to the same limiting case as the periodic open cylinder, namely $\Theta_\perp^{(\text{oo})}(x_\perp)$, see (2.56), which leads to a change in the sign where the two behaviours compete against each other.

without any additional divergence. Eventually we come to the strip contribution and, following our previous calculations, we use (2.31) to introduce

$$\delta\Psi^{(\text{oo})}(x_\parallel,\rho) = -\ln\frac{P_{\text{e}}^{(\text{oo})}(x_\parallel,\rho)}{P_{\text{o}}^{(\text{oo})}(x_\parallel,\rho)}. \tag{2.53}$$

Thus the scaling function for the surface tension reads

$$\Sigma_{\text{strip}}^{(\text{oo,a})}(x_\parallel,\rho) = \delta\Psi^{(\text{oo})}(x_\parallel,\rho). \tag{2.54}$$

For the antiperiodic strip contribution we thus find

$$\Psi^{(\text{oo,a})}(x_\parallel,\rho) = \Psi^{(\text{oo,p})}(x_\parallel,\rho) + \Sigma_{\text{strip}}^{(\text{oo,a})}(x_\parallel,\rho) = -\ln P_{\text{e}}^{(\text{oo})}(x_\parallel,\rho). \tag{2.55}$$

The total surface tension scaling function is shown in Figs. 6 and 7 for different values of $\rho$. For $\rho \leq 1$ we switch to the form with dominant perpendicular scaling variable $x_\perp$, where, nevertheless, the function vanishes for all $x_\perp$ as $\rho$ tends to zero.

Consequently, in the limit of thin films, the scaling functions for the open cylinder with periodic and antiperiodic BCs both converge to the well known scaling function of the open strip [40]

$$\Theta_\perp^{(\text{oo})}(x_\perp) = -\frac{1}{4\pi}\int\limits_{-\infty}^{\infty} d\Phi_\perp \, \ln\left[1 + \frac{\Gamma_\perp - x_\perp}{\Gamma_\perp + x_\perp}e^{-2\Gamma_\perp}\right], \tag{2.56}$$

with $\Gamma_\perp = \sqrt{x_\perp^2 + \Phi_\perp^2}$, which is the correct limit of (2.33) and (2.40) for $\rho \to 0$. Note that in this limit we have switched to the perpendicular representation, with $\Phi_\perp = \rho\Phi$. Finally, we can collect all results for (2.40); the corresponding scaling functions are shown in Figs. 8 and 9 for the $\rho \geq 1$ and $\rho \leq 1$, respectively.

# 3 Boundary fields and (+o) BCs

For the influence of BFs we switch back to the finite lattice. Applying local reduced BFs $h_m^{(l)}$ and $h_m^{(r)}$ to the spins on the left and the right side of the system, respectively, modifies the Hamiltonian according to

$$\mathcal{H}^{(\alpha,\mathrm{p})}(\{h\}) = \mathcal{H}^{(\mathrm{oo},\mathrm{p})} + \mathcal{H}^{(\alpha)}(\{h\}), \tag{3.1a}$$

with the boundary field Hamiltonian

$$\mathcal{H}^{(\alpha)}(\{h\}) = -\sum_{m=1}^{M} \left( h_m^{(l)} \sigma_{1,m} + h_m^{(r)} \sigma_{L,m} \right), \tag{3.1b}$$

where we allow the fields to vary depending on its position so we can dictate the BCs $\alpha$. This gives an additional term in both the singular and the non-singular part of the partition function, and following the same procedure as before, the latter one simply reads

$$Z_0^{(\alpha)}(\{h\}) = \prod_{m=1}^{M} \cosh h_m^{(l)} \cosh h_m^{(r)}. \tag{3.2a}$$

The singular part contributes as an additional factor within the sum over all spin configurations with

$$Z^{(\alpha)}(\{h\}) = \prod_{m=1}^{M} \left( 1 + \sigma_{1,m} z_m^{(l)} \right)\left( 1 + \sigma_{L,m} z_m^{(r)} \right) \tag{3.2b}$$

and $z_m^{(l/r)} = \tanh h_m^{(l/r)}$. Again we will assume homogeneous anisotropy, leaving us with the BF variables $z_m^{(l/r)} \equiv z_{l/r} \; \forall \, m$.

  We will start by applying only one boundary field to the cylinder and, following McCoy and Wu [50, chap. IV], we do so by adding a line of infinitely strong coupled spins either to the left or the right column. If we add a column to the left side of the system, i. e., at $\ell = 0$, the corresponding matrix reads

$$\mathcal{C}_{L+1}^{(+\mathrm{o})}(z_\mathrm{h};\varphi_m) = \left(\begin{array}{cc|cc} a_0 & 0 & & \\ 0 & a_1 & -1 & \\ \hline & -1 & & \\ & & & \mathcal{C}_{L-1}^{(0)} \\ \end{array}\right), \tag{3.3a}$$

with the boundary entries

$$a_0 = \frac{t_-}{z}\mu_0^+(\varphi_m), \qquad\qquad a_1 = \frac{t_-}{z}\left[\mu^+(\varphi_m) - z_\mathrm{h}^2 \mu_0^-(\varphi_m)\right], \tag{3.3b}$$

for a homogeneous boundary field $z_l = z_\mathrm{h}$, where we have introduced the boundary terms

$$\mu_0^\pm(\varphi_m) \equiv \mu^\pm(\varphi_m)\big|_{t=0} = \cos\varphi_m \mp 1, \tag{3.4}$$

as $t = 0$ corresponds to an infinitely strong coupling in parallel direction. Thus we have to apply (2.4a) twice and obtain

$$\eta_\pm^{(+\mathrm{o})}(z_\mathrm{h};\varphi_m) = \mp\left[ \mathrm{e}^{\mp\gamma_m} + \frac{t_-}{z}\left(\mu^+(\varphi_m) - z_\mathrm{h}^2 \mu_0^-(\varphi_m)\right)\right] \tag{3.5a}$$

$$= \eta_\pm^{(\mathrm{oo})}(\varphi_m)\,\eta_\pm^{(\mathrm{h})}(z_\mathrm{h};\varphi_m), \tag{3.5b}$$

with the boundary field contribution

$$\eta_{\pm}^{(\mathrm{h})}(z_{\mathrm{h}}; \varphi_m) \equiv 1 \mp z_{\mathrm{h}}^2 \frac{t_-}{z} \frac{\mu_0^-(\varphi_m)}{\eta_{\pm}^{(\mathrm{oo})}(\varphi_m)}. \tag{3.6}$$

Thus the final determinant using (2.10) reads

$$\det \mathcal{C}_{L+1}^{(+\mathrm{o})}(z_{\mathrm{h}}; \varphi_m) = \frac{t_-}{z} \mu_0^+(\varphi_m) e^{L\gamma_m} \frac{\eta_+^{(\mathrm{oo})}(\varphi_m) \eta_+^{(\mathrm{h})}(z_l; \varphi_m)}{2 \sinh \gamma_m} \times$$
$$\times \left[ 1 + \frac{\eta_-^{(\mathrm{oo})}(\varphi_m) \eta_-^{(\mathrm{h})}(z_l; \varphi_m)}{\eta_+^{(\mathrm{oo})}(\varphi_m) \eta_+^{(\mathrm{h})}(z_l; \varphi_m)} e^{-2L\gamma_m} \right]. \tag{3.7}$$

If we instead add an additional column to the right side of the system, i. e., at $\ell = L + 1$, we find the matrix

$$\mathcal{C}_{L+1}^{(\mathrm{o}+)}(z_{\mathrm{h}}; \varphi_m) = \begin{pmatrix} a_1 & -1 & & \\ -1 & & & \\ & & \mathcal{C}_{L-1}^{(0)} & \\ & & & b_L \\ & & b_L & a_{L+1} \end{pmatrix}, \tag{3.8a}$$

with $z_r = z_{\mathrm{h}}$ and with the matrix elements

$$a_1 = \frac{t_-}{z} \mu^+(\varphi_m) \qquad\qquad a_{L+1} = \frac{t_-}{z} \left[ \mu_0^+(\varphi_m) - z_{\mathrm{h}}^2 \mu^-(\varphi_m) \right], \tag{3.8b}$$

$$b_L = -\frac{z_{\mathrm{h}}}{z}. \tag{3.8c}$$

Here we have to apply both recursion formulas (2.4a) and (2.4b) and thus the calculation is a little bit nasty as it includes some additional symmetry considerations; it is shown in detail in Appendix B. The final result then reads, of course,

$$\det \mathcal{C}_{L+1}^{(+\mathrm{o})}(z_{\mathrm{h}}; \varphi_m) = \det \mathcal{C}_{L+1}^{(\mathrm{o}+)}(z_{\mathrm{h}}; \varphi_m), \tag{3.9}$$

because of the symmetry of the system with respect to reversing the perpendicular direction. Finally the partition function reads

$$\frac{Z^{(+\mathrm{o},\mathrm{p})}}{Z_0^{(+\mathrm{o},\mathrm{p})}} = \frac{1}{2} \left( \frac{2z}{1+t_+} \right)^{\frac{LM}{2}} \left( -\frac{2z}{t_-} \right)^{\frac{M}{2}} \prod_{\substack{0 < m < 2M \\ m\,\mathrm{odd}}} \det^{\frac{1}{2}} \mathcal{C}_{L+1}^{(+\mathrm{o})}(z_{\mathrm{h}}; \varphi_m). \tag{3.10}$$

Note that we only assume periodic BCs in the parallel direction, since the BFs and the antiperiodic BCs in combination need a more precise survey.

**Surface field contribution and scaling limit**

The previous calculation shows that the additional surface field can be easily separated as a correction to the open surfaces. Additionally, we obtain a factor $-\mu_0^+(\varphi)$, which corresponds directly to the additional line of infinitely-strong coupled spins. We may calculate the respective contribution to the partition function for the periodic case exactly as

$$\prod_{\substack{0 < m < 2M \\ m\,\mathrm{odd}}} \left[ \frac{t_-}{z} (\cos \varphi_m - 1) \right]^{\frac{1}{2}} = 2 \left( -\frac{t_-}{2z} \right)^{\frac{M}{2}}. \tag{3.11}$$

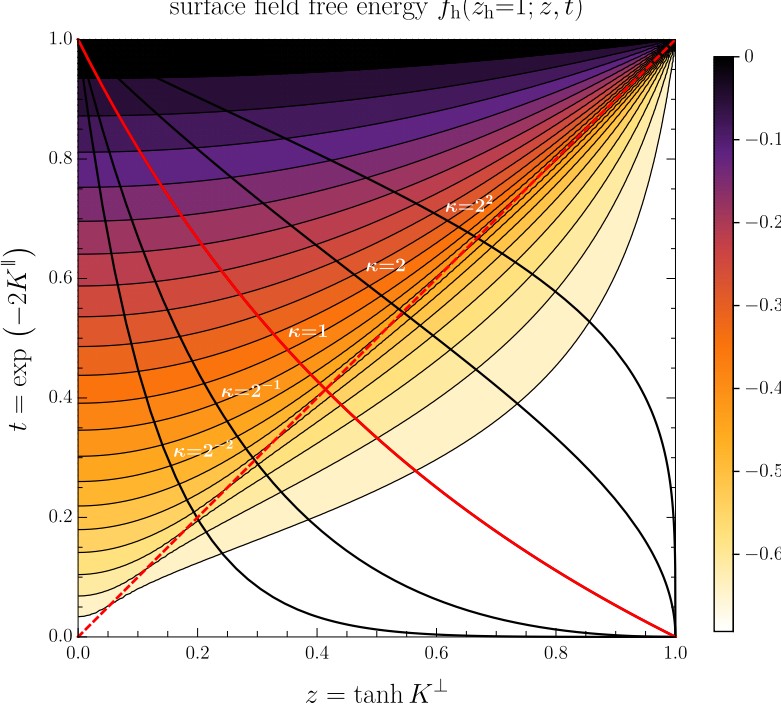

surface field free energy $f_\mathrm{h}(z_\mathrm{h}{=}1; z, t)$

Figure 10: Surface field free energy density $f_\mathrm{h}(z_\mathrm{h}{=}1; z, t)$, see (3.12). The dashed line marks criticality for arbitrary anisotropy $\kappa = K^\perp/K^\parallel$. For fixed anisotropy the two couplings are connected by $t = (z^*)^\kappa$ and the black lines mark the run of the related curves, where the isotropic case is shown in red.

The field contribution to the free energy in the thermodynamic limit is again given by the integral over the corresponding contribution $\eta_+^{(\mathrm{h})}(z_\mathrm{h}; \varphi)$ as

$$f_\mathrm{h}(z_\mathrm{h}; z, t) = -\frac{1}{4\pi} \int\limits_0^{2\pi} \mathrm{d}\varphi \, \ln \eta_+^{(\mathrm{h})}(z_\mathrm{h}; \varphi) \tag{3.12}$$

and is depicted in Fig. 10. Its critical value for the isotropic case and infinitely strong field can be calculated exactly in terms of Catalan's constant $G$ and the critical coupling $z_\mathrm{c}$, and reads

$$f_\mathrm{h}(z_\mathrm{h}{=}1; z_\mathrm{c}, z_\mathrm{c}) = \frac{1}{2} \ln \frac{z_\mathrm{c}}{2} + \frac{G}{\pi}. \tag{3.13}$$

For the scaling function it is necessary to consider the case of an infinitely strong surface field, i.e., $z_\mathrm{h} = 1$, otherwise its effect would vanish in the desired limit, and more important, we could not use our likewise simple methods. A corresponding scaling theory for the strip geometry, i.e., $M \to \infty$ for varying BFs was given in [41]. Such a field breaks the $Z_2$-symmetry of the system and the scaling function goes to zero in both the unordered and the ordered phase in contrast to the scaling functions for the fully periodic torus or the periodic open cylinder. Indeed this reflects the duality of the two present BCs; under a duality transformation, an open boundary transforms into a symmetry-breaking one and vice versa, while additionally the high- and the low-temperature phase interchange. Nevertheless, this symmetry is only exact for at least in one direction infinitely large systems, i.e., the thin film limit, as otherwise the corrections do not vanish. The next two blocks we need to calculate in the scaling limit thus are the two $\eta_\pm^{(\mathrm{h})}(z_\mathrm{h}{=}1; \varphi)$.

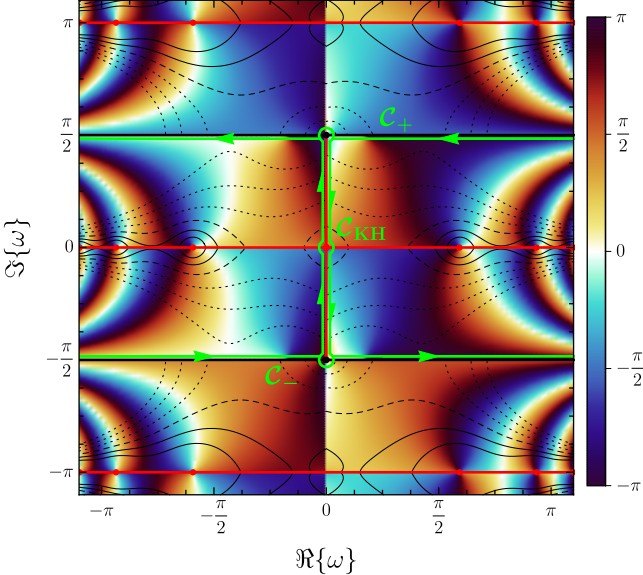

Figure 11: Complex structure of the contour integrand (3.22) in the hyperbolic $\omega$-plane for the odd sum over the $\ln\left(2\Gamma_m\right) - \ln\left(\Phi_m^2\right)/2$, together with the contour $\mathcal{C} = \mathcal{C}_+ + \mathcal{C}_- + \mathcal{C}_{\mathrm{KH}}$. Since this is the correction of the surface field to the open boundaries, the some kind of *regular* part from the contours $\mathcal{C}_\pm$ is zero and only the nontrivial keyhole contour $\mathcal{C}_{\mathrm{KH}}$ contributes. Note that there is no explicit dependency on $x_\parallel$ and thus we do not need to distinguish between the ordered and unordered phase. For the colour coding see Fig. 2.

We start with a series expansion of $\eta_+^{(\mathrm{h})}(z_\mathrm{h}{=}1; \varphi)$ at criticality and find not only a logarithmic divergence in $M$ but also a constant term which still contains the anisotropy variable $r_\xi$,

$$\ln \eta_+^{(\mathrm{h})}(z_\mathrm{h}{=}1; \varphi)\Big|_{z=t} \simeq \ln\left(2a_\xi M\right) - \frac{1}{2}\ln\left(\Phi^2\right), \tag{3.14}$$

with the inverse critical coupling

$$a_\xi \equiv \frac{r_\xi}{\sqrt{1 + r_\xi^2} - 1}. \tag{3.15}$$

We can use the critical value to regularise its non-critical counterpart, as all the diverging and non-universal contributions cancel each other out. Then we need to calculate the sum over (3.14) in a regularised form; therefore we can use the zeta-regularisation. First we notice that the summation over the odd numbers is symmetric and thus we can write

$$\sum_{\substack{m=-\infty \\ m\,\mathrm{odd}}}^{\infty} \ln \eta_+^{(\mathrm{h})}(z_\mathrm{h}{=}1; \varphi_m)\Big|_{z=t} = 2 \sum_{\substack{m>0 \\ m\,\mathrm{odd}}}^{\infty} \left[\ln\left(2a_\xi M\right) - \ln \Phi_m\right] \tag{3.16}$$

by assuming $\Phi > 0$. The zeta-regularised sum over the odd numbers to the power of $(-s)$ reads

$$\sum_{m=1}^{\infty} (2m-1)^{-s} = \frac{2^s - 1}{(2\pi)^s}\zeta(s), \tag{3.17}$$

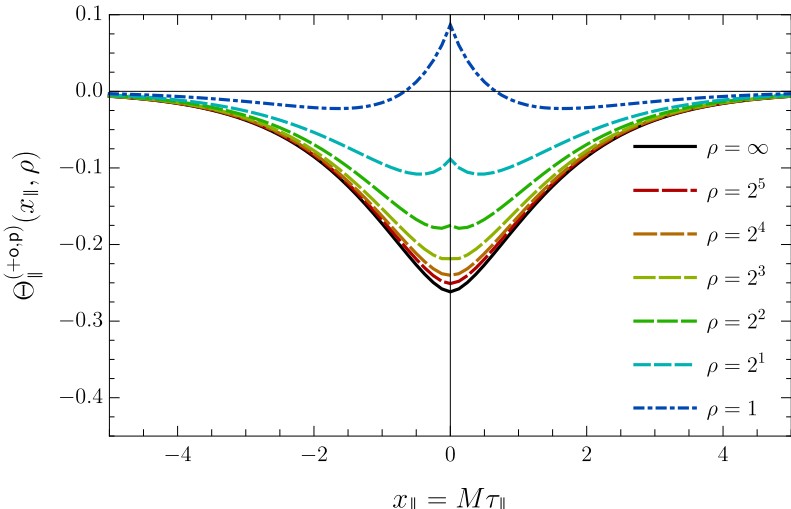

Figure 12: Scaling function $\Theta_{\parallel}^{(+o,p)}(x_{\parallel},\rho)$ of the free energy for different values of aspect ratios $\rho \geq 1$. For $\rho \neq \infty$ there is a kink at $x_{\delta} = 0$ present and, in contrast to the periodic systems we discussed before, the scaling function converges to zero for all aspect ratios in both the ordered and the unordered phase due to the unambiguously broken $Z_2$-symmetry.

with the Riemann zeta function $\zeta(s)$. For the constant part we take $s \to 0$ and find

$$2 \sum_{\substack{m>0 \\ m\,\text{odd}}}^{\infty} \ln\left(2a_{\xi}M\right) = 0, \tag{3.18}$$

that is, these terms do not contribute to the scaling limit, as they cancel out with the thermodynamic limit. The logarithmic term can be calculated by a derivative of (3.17) as

$$2 \sum_{\substack{m>0 \\ m\,\text{odd}}}^{\infty} \ln\Phi_m = -2 \lim_{s\to 0} \frac{\mathrm{d}}{\mathrm{d}s}\left[\frac{2^s-1}{(2\pi)^s}\,\zeta(s)\right] = \ln 2. \tag{3.19}$$

Now we can regularise the temperature depending form of $\eta_+^{(h)}(z_{\mathrm{h}}{=}1;\varphi)$ with its critical value and find

$$\ln \frac{\eta_+^{(h)}(z_{\mathrm{h}}{=}1;\varphi)}{\eta_+^{(h)}(z_{\mathrm{h}}{=}1;\varphi)\big|_{z=t}} \simeq -\ln \frac{\Gamma + x_{\parallel}}{|\Phi|}. \tag{3.20}$$

Since we already calculated the sum over $\ln\left[(\Gamma_m + x_{\parallel})/(2\Gamma_m)\right]$ in section 2 for the open cylinder, we decompose (3.20) into

$$-\ln \frac{\Gamma + x_{\parallel}}{|\Phi|} = \ln(2\Gamma) - \frac{1}{2}\ln\left(\Phi^2\right) - \ln \frac{\Gamma + x_{\parallel}}{2\Gamma}, \tag{3.21}$$

and thus we only have to calculate the sum over the first two summands. The terms are independent of whether we focus on the ordered or the unordered phase and thus we only have to do the calculation once, which is especially interesting as the integrals over the shifted contours $\mathcal{C}_+$ and $\mathcal{C}_-$ vanish and only the contribution of the keyhole contour $\mathcal{C}_{\mathrm{KH}}$ around the logarithmic cut is relevant, see Fig. 11. The logarithmic divergence cancels out again due to

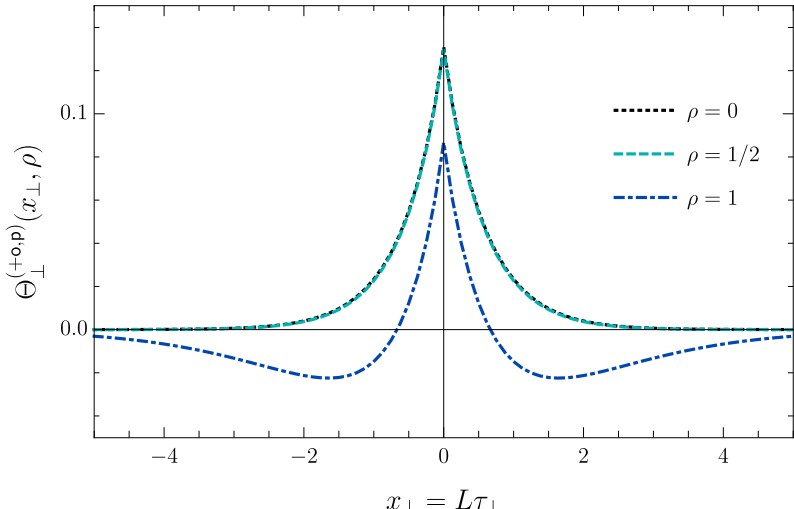

Figure 13: Scaling function $\Theta_{\perp}^{(+o,p)}(x_{\perp}, \rho)$ for different values of the aspect ratio $\rho \leq 1$. The scaling function converges rather fast agains its limiting case (3.29) and changes its sign for $\rho \approx 1$ depending on the temperature.

the symmetries around the origin and we find

$$\frac{1}{4i\pi} \oint_{\mathcal{C}_{\text{KH}}} d\omega \, |x_{\parallel}| \cosh \omega \left[ \ln\left(2|x_{\parallel}| \cosh \omega\right) - \frac{1}{2} \ln\left(x_{\parallel}^2 \sinh^2 \omega\right) \right] \mathcal{K}_{\text{o}}^{\pm}(|x_{\parallel}| \sinh \omega) \tag{3.22}$$

$$= \frac{1}{2} \ln\left[ 1 + e^{-|x_{\parallel}|} \right],$$

which leaves us with

$$\Theta_{\text{s}}^{(h,p)}(x_{\parallel}) = \frac{1}{2} \ln\left[ 1 + e^{-|x_{\parallel}|} \right] - \Theta_{\text{s}}^{(oo,p)}(x_{\parallel}) \tag{3.23}$$

for the surface field scaling function.

For the strip residual free energy, we only need to calculate the fraction

$$\frac{\eta_{-}^{(h)}(z_{\text{h}}=1; \varphi)}{\eta_{+}^{(h)}(z_{\text{h}}=1; \varphi)} \simeq \frac{\Gamma + x_{\parallel}}{\Gamma - x_{\parallel}}, \tag{3.24}$$

which regularises itself, and thus we can write down the corresponding product

$$P_{\text{e/o}}^{(+o)}(x_{\parallel}, \rho) = \prod_{\substack{m > 0 \\ m \, \text{even/odd}}}^{\infty} \left[ 1 + e^{-2\rho\Gamma_m} \right], \tag{3.25}$$

where the contributions from the open boundaries and the surface field contribute as the inverse of each other and thus cancel out. The strip scaling function then is, analogous to (2.32), simply

$$\Psi^{(+o,p)}(x_{\parallel}, \rho) = -\ln P_{\text{o}}^{(+o)}(x_{\parallel}, \rho) \tag{3.26}$$

and the complete scaling function reads

$$\rho \, \Theta_{\parallel}^{(+o,p)}(x_{\parallel}, \rho) = \rho \, \Theta_{\text{b}}^{(p)}(x_{\parallel}) + \Theta_{\text{s}}^{(oo,p)}(x_{\parallel}) + \Theta_{\text{s}}^{(h,p)}(x_{\parallel}) + \Psi^{(+o,p)}(x_{\parallel}, \rho) \tag{3.27}$$

and is depicted in Figs. 12 and 13. We can see that for systems with sufficiently small $\rho$ there is a kink at $x_\delta = 0$ and, depending on the temperature and the aspect ratio, the scaling function changes it sign. Subsequently we can identify the two limiting cases for $\rho \to 0$ and $\rho \to \infty$; the latter one is again dominated by the periodicity and thus we find

$$\Theta_\parallel^{(+o,p)}(x_\parallel, \rho \to \infty) = \Theta_b^{(p)}(x_\parallel), \tag{3.28}$$

while the former one is approached rather fast, almost already for $\rho = 1/2$, and reads

$$\Theta_\perp^{(+o)}(x_\perp) \equiv \Theta_\perp^{(+o,p)}(x_\perp, \rho \to 0) = -\frac{1}{4\pi} \int\limits_{-\infty}^{\infty} d\Phi_\perp \ln\left[1 - e^{-2\Gamma_\perp}\right], \tag{3.29}$$

cf. (2.56). For growing aspect ratio the scaling function changes its sign from an all negative to an all positive function with a temperature-dependent change for $\rho \approx 1$.

## 4 Staggered fields ($\uparrow\downarrow$ o) at one surface

Instead of a homogeneous field, we may impose a staggered field on one boundary, which is believed to be equivalent to Dirichlet and open boundaries in the thermodynamic limit [33]. The other boundary is still open. While this combination of BCs correspond to the original

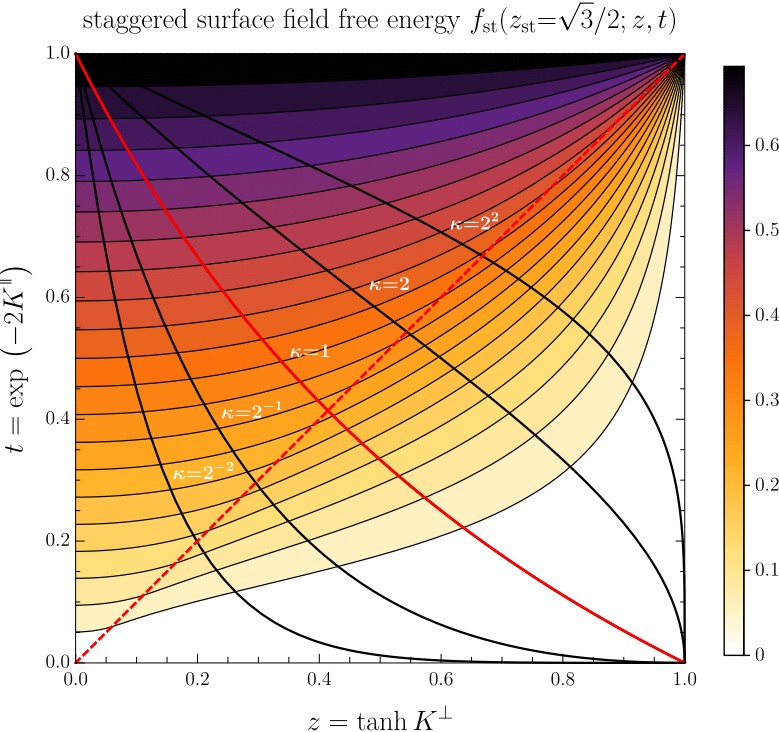

Figure 14: Surface free energy density of a staggered field $f_{st}(z_{st}; z, t)$, see (4.5). The dashed line marks criticality for arbitrary anisotropy $\kappa = K^\perp/K^\parallel$. For fixed anisotropy the two couplings are connected by $t = (z^*)^\kappa$ and the black lines mark the run of the related curves, where the isotropic case is shown in red. Note that we have chosen $z_{st} = \sqrt{3}/2$ as it has a maximal value of $\ln 2$. For $z_{st} = 1$ the density diverges logarithmically for $t \to 1$.

setup of Brascamp and Kunz [39], the staggered field at one boundary is also often denoted Brascamp-Kunz BC in the literature. We again couple the spins in the additional line infinitely strong, but this time with the opposite sign, i. e., $z_{L+1}^{\parallel} = -1$. As the other direction is periodic, $M$ must be even. This changes every $\mu_0^{\pm}(\varphi) \to \mu_0^{\mp}(\varphi)$ in (3.3a), and thus we have to deal with the matrix

$$
\mathcal{C}_{L+1}^{(\uparrow\downarrow o,p)}(z_{st}; \varphi_m) =
\begin{pmatrix}
\begin{matrix} a_0 & 0 \\ 0 & a_1 \end{matrix} & -1 & \\
-1 & & \\
& & \mathcal{C}_{L-1}^{(0)}
\end{pmatrix},
\tag{4.1a}
$$

with staggered field $z_{st} = \tanh h_{st}$ and with the boundary entries

$$
a_0 = \frac{t_-}{z}\mu_0^-(\varphi_m), \qquad\qquad a_1 = \frac{t_-}{z}\left[\mu^+(\varphi_m) - z_{st}^2\mu_0^+(\varphi_m)\right].
\tag{4.1b}
$$

Thus we can calculated the contribution for the staggered surface field as

$$
\eta_{\pm}^{(\uparrow\downarrow o)}(z_{st}; \varphi_m) = \mp\left(e^{\mp\gamma_m} + \frac{t_-}{z}\left[\mu^+(\varphi_m) - z_{st}^2\mu_0^+(\varphi_m)\right]\right)
\tag{4.2a}
$$

$$
= \eta_{\pm}^{(oo)}(\varphi_m)\,\eta_{\pm}^{(\uparrow\downarrow)}(z_{st}; \varphi),
\tag{4.2b}
$$

such that

$$
\eta_{\pm}^{(\uparrow\downarrow)}(z_{st}; \varphi) \equiv 1 \mp z_{st}^2\frac{t_-}{z}\frac{\mu_0^+(\varphi_m)}{\eta_{\pm}^{(oo)}(\varphi_m)}
\tag{4.3}
$$

for a staggered field. Additionally we find the contribution of the infinitely strong coupled line of spins to be

$$
\prod_{\substack{0 < m < 2M \\ m\,\text{odd}}} \left[\frac{t_-}{z}(1 + \cos\varphi_m)\right]^{\frac{1}{2}} = 2\left(-\frac{t_-}{2z}\right)^{\frac{M}{2}},
\tag{4.4}
$$

which is equal to the one for a homogeneous field, as it gives the correction to the surface and a factor of two due to the $Z_2$ symmetry of this macro spin. The corresponding free energy contribution of a staggered surface field is given by

$$
f_{st}(z, t; z_{st}) = -\frac{1}{4\pi}\int_0^{2\pi} d\varphi \ln \eta_+^{(\uparrow\downarrow)}(z_{st}; \varphi),
\tag{4.5}
$$

see Fig. 14.

For the scaling function we choose $z_{st} = 1$ and find the scaling form of $\eta_{\pm}^{(\uparrow\downarrow)}(z_{st} = 1; \varphi)$ as

$$
\eta_{\pm}^{(\uparrow\downarrow)}(z_{st} = 1; \varphi) \simeq 1.
\tag{4.6}
$$

Thus we see that both finite-size scaling function of the system with open BCs as well as the one with an infinitely-strong staggered field are the same,

$$
\Theta_{\parallel}^{(\uparrow\downarrow o,p)}(x_{\parallel}, \rho) \equiv \Theta_{\parallel}^{(oo,p)}(x_{\parallel}, \rho).
\tag{4.7}
$$

Numerically, i. e., for the finite systems, we see that they converge to the same scaling function with growing system size from opposite directions. We will later resume to this point, when we examine the doubly staggered BCs ($\uparrow\downarrow\uparrow\downarrow$) in section 6.

# 5 Symmetry-breaking BCs at both surfaces

For symmetry-breaking boundary conditions at both surfaces we follow the same procedure as for one boundary field, but we couple the additional infinitely-strong coupled line to both the first and the last row, introducing the two boundary fields $z_l$ and $z_r$ and again forming a torus. It is easy to see that a $b_k$ on a boundary is either 0 if a perpendicular coupling is cut open, or $\pm 1$ if a row in parallel direction is coupled infinitely strong either ferro- or anti-ferromagnetic. The difference between the (++)- and the (+−) boundary conditions is solely yielded by the choice between periodic and antiperiodic continuity in the perpendicular direction, respectively. This makes it necessary to switch back to the formulation of the torus with the sum over four Pfaffians. The final matrix reads

$$
\mathcal{C}^{(++)}_{L+1}(z_l, z_r; \varphi_m) =
\begin{pmatrix}
a_1 & -1 & & & b_{L+1} \\
\hline
-1 & & & & \\
& & \mathcal{C}^{(0)}_{L-1} & & \\
& & & & b_L \\
\hline
b_{L+1} & & & b_L & a_{L+1}
\end{pmatrix},
\tag{5.1a}
$$

with

$$
a_1 = \frac{t_-}{z}\left[\mu^+(\varphi_m) - z_l^2 \mu_0^-(\varphi_m)\right], \qquad\qquad b_{L+1} = 0,
\tag{5.1b}
$$

$$
a_{L+1} = \frac{t_-}{z}\left[\mu_0^+(\varphi_m) - z_r^2 \mu^-(\varphi_m)\right], \qquad\qquad b_L = -\frac{z_r}{z}.
\tag{5.1c}
$$

A detailed calculation of the determinant can be found in Appendix B, and the final result reads

$$
\det \mathcal{C}^{(++)}_{L+1}(z_l, z_r; \varphi_m) = \frac{t_-}{z}\mu_0^+(\varphi_m)\, e^{L\gamma_m} \frac{\eta_+^{(\text{oo})}(\varphi_m)\,\eta_+^{(\text{h})}(z_l; \varphi_m)\,\eta_+^{(\text{h})}(z_r; \varphi_m)}{2\sinh\gamma_m} \times
$$
$$
\times \left[1 + \frac{\eta_-^{(\text{oo})}(\varphi_m)\,\eta_-^{(\text{h})}(z_l; \varphi_m)\,\eta_-^{(\text{h})}(z_r; \varphi_m)}{\eta_+^{(\text{oo})}(\varphi_m)\,\eta_+^{(\text{h})}(z_l; \varphi_m)\,\eta_+^{(\text{h})}(z_r; \varphi_m)}\, e^{-2L\gamma_m}\right].
\tag{5.2}
$$

Following the procedure we used for the torus, we find the singular part of the partition function to be

$$
\frac{Z^{(+\pm,\text{p})}}{Z_0^{(+\pm,\text{p})}} = \frac{1}{2}\left(\frac{2z}{1+t_+}\right)^{\frac{LM}{2}}\left(-\frac{2z}{t_-}\right)^{\frac{M}{2}} \times
$$
$$
\times \left[\prod_{\substack{0 < m < 2M \\ m\,\text{odd}}} \det^{\frac{1}{2}}\mathcal{C}^{(++)}_{L+1}(\varphi_m) \pm \prod_{\substack{0 \le m < 2M \\ m\,\text{even}}} \det^{\frac{1}{2}}\mathcal{C}^{(++)}_{L+1}(\varphi_m)\right],
\tag{5.3}
$$

as due to the infinitely strong coupled line of spins each pair of Pfaffians is identical and we only have an even and an odd contribution left.

**The alternating contribution**

For the scaling function we first set $z_l = z_r = z_{\text{h}}$ and then factorise the odd contribution from the partition function like we did for the torus, see section I.4, leaving us with the odd part, for which we already know all scaling functions but an alternating part from the BF contributions, which we will discuss now. Therefore we will proceed in the same manner as we did for the surface tension of the antiperiodic open cylinder; we take the square root by

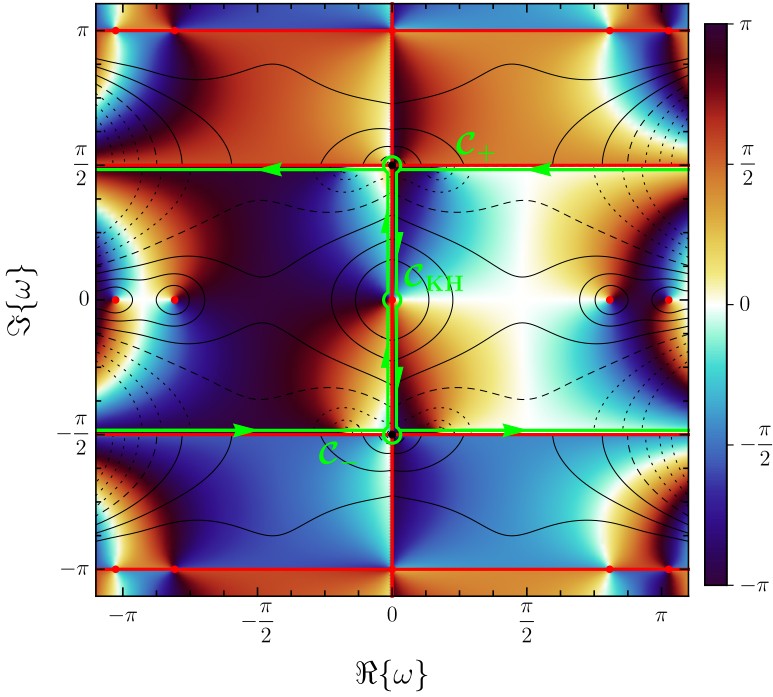

Figure 15: Complex structure of the contour integrand in the hyperbolic $\omega$-plane for the alternating sum over $\ln[2\Gamma_m] - \ln[\Phi_m^2]/2$, together with the contour $\mathcal{C} = \mathcal{C}_+ + \mathcal{C}_- + \mathcal{C}_{\mathrm{KH}}$. The contributions from the contours $\mathcal{C}_\pm$ vanish due to their symmetry and only the non-trivial keyhole contour $\mathcal{C}_{\mathrm{KH}}$ is relevant for the correction to the behaviour of the open surfaces. For the colour coding see Fig. 2.

using the symmetry of the eigenvalues $\gamma_m$ as every factor but the two symmetry points at $\varphi = 0$ and $\varphi = \pi$ appear exactly twice. Its values at those points are

$$\det \mathcal{C}_{L+1}^{(++)}(z_{\mathrm{h}}, z_{\mathrm{h}}; \varphi = 0) = -2z t_- \left(\frac{z}{t}\right)^L \left(\frac{z_{\mathrm{h}}}{z}\right)^4, \tag{5.4a}$$

$$\det \mathcal{C}_{L+1}^{(++)}(z_{\mathrm{h}}, z_{\mathrm{h}}; \varphi = \pi) = -\frac{2t_-}{z} \left(\frac{1}{zt}\right)^L. \tag{5.4b}$$

For convenience we change to the logarithm and find for the alternating sum

$$\sum_{m=0}^{2M-1} (-1)^m \ln \det^{\frac{1}{2}} \mathcal{C}_{L+1}^{(++)}(z_{\mathrm{h}}, z_{\mathrm{h}}; \varphi_m)$$

$$= \ln\left[-2t_- \frac{z_{\mathrm{h}}^2}{z^2}\right] - L \ln t + \sum_{m=1}^{M-1} (-1)^m \ln \det \mathcal{C}_{L+1}^{(++)}(z_{\mathrm{h}}, z_{\mathrm{h}}; \varphi_m), \tag{5.5}$$

so we just have to handle sums instead of products. We start with the bulk contribution, which we identify as all terms linear in $L$ as

$$-L \ln t + L \sum_{m=1}^{M-1} (-1)^m \gamma_m = -L \ln \frac{t}{z} H(t-z) + \frac{L}{2} \sum_{m=1}^{2M} (-1)^m \gamma_m, \tag{5.6}$$

where we reformulated the sum in such a way that we can identify the scaling limit we have already calculated for the torus. The additional term is slightly different from (2.41), nevertheless its scaling form is given by

$$-L \ln \frac{t}{z} H(t-z) \simeq -\rho \, x_\parallel H(x_\parallel), \tag{5.7}$$

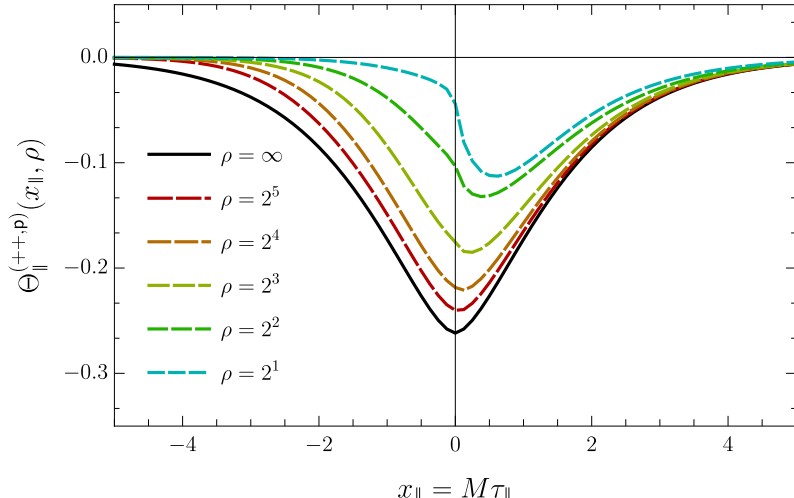

Figure 16: Scaling function $\Theta_\parallel^{(++,\text{p})}(x_\parallel, \rho)$ of the free energy for different values of aspect ratios $\rho \geq 2$. The scaling function converges against the limiting case of a periodic thin film $\Theta_\text{b}(x_\parallel)$ for growing $\rho$. At about $\rho \approx 2$ the behaviour starts to change dramatically, see Fig. 17.

that is, the linear divergence is in contrast to (2.44) in the unordered phase.

For the surface contribution we find

$$
\ln\left(-\frac{2t_-}{z^2}\right) + \sum_{m=1}^{M-1}(-1)^m \ln\left[\frac{t_-}{z}\mu_0^+(\varphi_m)\,\eta_+^{(\text{h})}(z_\text{h}{=}1;\varphi_m)^2\right]
$$
$$
= \ln M - \ln z + 2\sum_{m=1}^{M-1}(-1)^m \ln \eta_+^{(\text{h})}(z_\text{h}{=}1;\varphi_m),
$$
(5.8)

where we used

$$
\sum_{m=1}^{M-1}(-1)^m \ln\mu_0^+(\varphi_m) = \ln\frac{M}{2}.
$$
(5.9)

To calculate the scaling limit we use the same fruitful approach as for one surface field, i. e., the zeta regularisation for the critical value, but this time we need to do it for an alternating sum. We use (3.14) and and write for the scaling form of the critical case

$$
\sum_{m=1}^{M-1}(-1)^m \ln\eta_+^{(\text{h})}(z_\text{h}{=}1;\varphi_m)\Big|_{z=t} \simeq \sum_{m=1}^{\infty}(-1)^m\Big[\ln\big(2a_\xi M\big) - \ln\Phi_m\Big].
$$
(5.10)

The $\ln M$ is just a constant, and we handle it as an independent term from some kind of cutoff, and additionally we have

$$
-\ln z \simeq \ln a_\xi.
$$
(5.11)

The zeta regularisation of the alternating sum reads

$$
\sum_{m=1}^{\infty}(-1)^m m^{-s} = \frac{2-2^s}{(2\pi)^s}\zeta(s),
$$
(5.12)

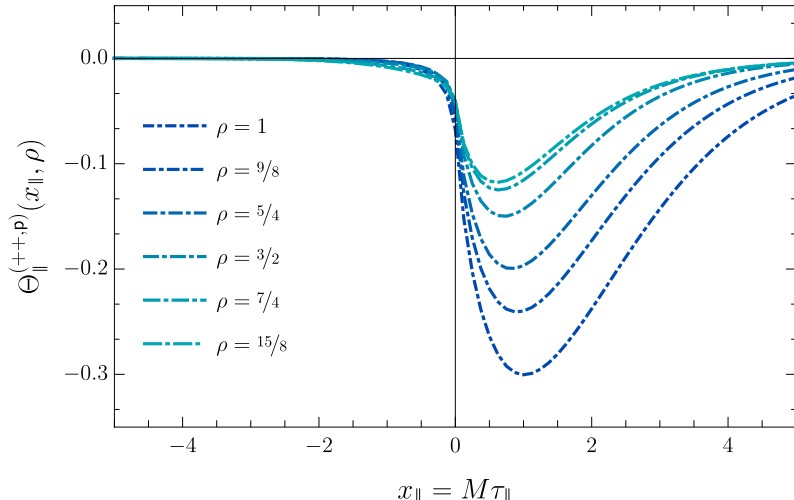

Figure 17: Scaling function $\Theta_\parallel^{(++,\mathrm{p})}(x_\parallel,\rho)$ for different values of the aspect ratio $1 \le \rho \le 2$. The scaling function converges rather fast agains its limiting case (5.21), and its minimum is minimal for $\rho \approx 2$.

and thus we find for the constant terms with $s = 0$

$$\sum_{m=1}^{\infty}(-1)^m \ln(2a_\xi M) = -\frac{1}{2}\ln\left(2a_\xi M\right). \tag{5.13}$$

For the $\Phi$-dependent term we need to calculate the derivative with respect to $s$ again and afterwards let $s \to 0$, thus the alternating sum reads in zeta regularised form

$$-\sum_{m=1}^{\infty}(-1)^m \ln \Phi_m = \lim_{s\to 0}\frac{\mathrm{d}}{\mathrm{d}s}\left[\frac{2^s-2}{(2\pi)^s}\zeta(s)\right] = \frac{\ln 2}{2}. \tag{5.14}$$

Combining all these terms leads to

$$\ln M - \ln z + 2\sum_{m=1}^{M-1}(-1)^m \ln \eta_+^{(\mathrm{h})}(z_{\mathrm{h}}=1;\varphi_m) \simeq 2\sum_{m=1}^{\infty}(-1)^m \ln\frac{\Gamma_m+x_\parallel}{|\Phi_m|} \tag{5.15}$$

as all other terms cancel out, i. e., the additional terms for $\varphi = 0$ and $\varphi = \pi$ together correspond in the scaling limit of the critical value of the boundary contribution to this alternating sum.

Fortunately, for the surface tension we already calculated the contribution to the alternating sum for $\ln[(\Gamma_m + x_\parallel)/(2\Gamma_m)]$ in section 2. Therefore we are left with the alternating sum of $\ln(2\Gamma_m/|\Phi_m|) = \left[\ln(2\Gamma_m) - \ln(\Phi_m^2)/2\right]$ analogous to the calculation for one boundary field, which we will handle by the hyperbolic parametrisation and the corresponding contour integration. The complex structure is shown in Fig. 15. As before, the integration along $\mathcal{C}_\pm$ vanish, as their real part is an odd function and their imaginary part is zero and we are left with the keyhole contour integration around the logarithmic cut from $-\mathrm{i}\pi/2$ to $\mathrm{i}\pi/2$. We can proceed in the same manner as we did for the antiperiodic open cylinder in section 2 and split the integration kernel $\delta\mathcal{K}$ into its simple pole at $\omega = 0$ and the rest. A careful study of the two integrals gives the final result

$$\frac{1}{4\mathrm{i}\pi}\oint_{\mathcal{C}_{\mathrm{KH}}}\mathrm{d}\omega\,\delta\mathcal{K}(\Phi)\left[\ln(2\Gamma)-\frac{1}{2}\ln\left(\Phi^2\right)\right] = \ln\left[\frac{1}{x_\parallel}\tanh\frac{x_\parallel}{2}\right] \tag{5.16}$$

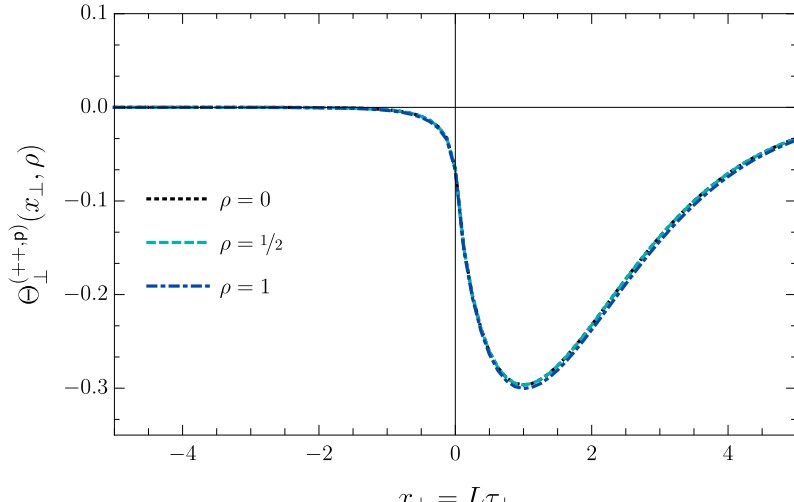

Figure 18: Scaling function $\Theta_\perp^{(++,\mathrm{p})}(x_\perp,\rho)$ of the free energy for different values of aspect ratios $\rho \le 1$. For $\rho < 1$, the scaling function converges extremely fast toward the thin film limit $\Theta_\perp^{(++)}(x_\perp)$.

and in the style of the alternating contributions we saw before, we thus end up with

$$\delta\Theta_{\mathrm{s}}^{(\mathrm{h})}(x_\parallel) = \frac{1}{2}\ln\left[\frac{1}{x_\parallel}\tanh\frac{x_\parallel}{2}\right] - \delta\Theta_{\mathrm{s}}^{(\mathrm{oo})}(x_\parallel) \tag{5.17}$$

as term from the surface field. For the strip contribution of two symmetry-breaking BFs, denoted as (hh), we simply find

$$P_{\mathrm{e/o}}^{(\mathrm{hh})}(x_\parallel,\rho) = \prod_{\substack{0 < m \\ m\,\mathrm{even/odd}}}^{\infty}\left[1 + \frac{\Gamma_m + x_\parallel}{\Gamma_m - x_\parallel}e^{-2\rho\Gamma_m}\right] \tag{5.18}$$

for the corresponding products, with $P_{\mathrm{e/o}}^{(\mathrm{hh})}(x_\parallel,\rho) = P_{\mathrm{e/o}}^{(\mathrm{oo})}(-x_\parallel,\rho)$, such that the associated scaling functions read

$$\Psi^{(\mathrm{hh},\mathrm{p})}(x_\parallel,\rho) = -\ln P_{\mathrm{o}}^{(\mathrm{hh})}(x_\parallel,\rho) = \Psi^{(\mathrm{oo},\mathrm{p})}(-x_\parallel,\rho), \tag{5.19a}$$

$$\delta\Psi^{(\mathrm{hh})}(x_\parallel,\rho) = -\ln\frac{P_{\mathrm{e}}^{(\mathrm{hh})}(x_\parallel,\rho)}{P_{\mathrm{o}}^{(\mathrm{hh})}(x_\parallel,\rho)} = \delta\Psi^{(\mathrm{oo})}(-x_\parallel,\rho). \tag{5.19b}$$

**Scaling functions**

Combining the results of the last section, we find the final form of the scaling function for both the symmetric and the antisymmetric symmetry-breaking BCs to be

$$\rho\,\Theta_\parallel^{(+\pm,\mathrm{p})}(x_\parallel,\rho) = \rho\,\Theta_{\mathrm{b}}^{(\mathrm{p})}(x_\parallel) + \Theta_{\mathrm{s}}^{(\mathrm{oo},\mathrm{p})}(x_\parallel) + 2\Theta_{\mathrm{s}}^{(\mathrm{h},\mathrm{p})}(x_\parallel) + \Psi^{(\mathrm{hh},\mathrm{p})}(x_\parallel,\rho)$$
$$+ \ln\left[1 \pm e^{-\rho\left[\delta\Theta_{\mathrm{b}}(x_\parallel)+x_\parallel H(x_\parallel)\right]-\delta\Theta_{\mathrm{s}}^{(\mathrm{oo})}(x_\parallel)-2\delta\Theta_{\mathrm{s}}^{(\mathrm{h})}(x_\parallel)-\delta\Psi^{(\mathrm{hh})}(x_\parallel,\rho)}\right], \tag{5.20}$$

which is remarkably similar to the formulas obtained by CFT [45, 52, 53] and agrees with the results of Evans and Stecki for the strip geometry [40].

Let us first take a closer look at the symmetric symmetry-breaking BCs, i.e., (++) boundaries. As we have seen before for several other systems, the scaling function converges toward

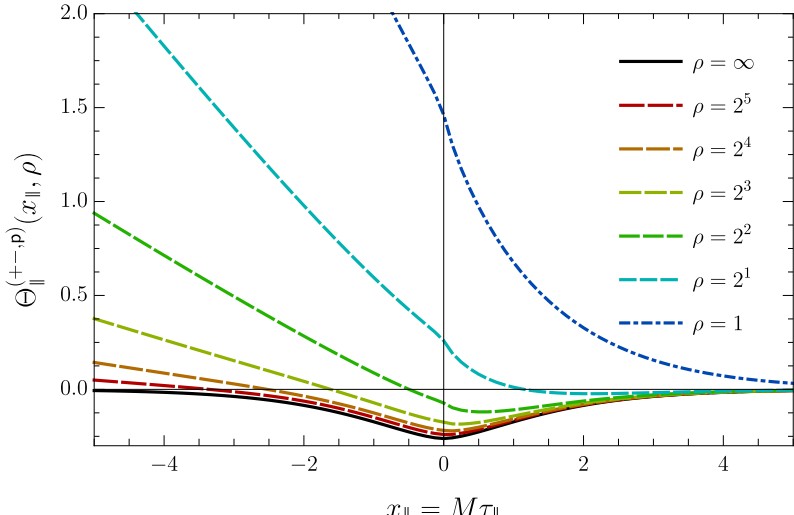

Figure 19: Scaling function $\Theta_{\parallel}^{(+-,\text{p})}(x_{\parallel},\rho)$ of the free energy for different values of aspect ratios $\rho \geq 1$. For growing $\rho$ the influence of the domain wall shrinks, until it vanishes completely for $\rho = \infty$, where the scaling functions approaches its limiting form $\Theta_{\text{b}}^{(\text{p})}(x_{\parallel})$.

$\Theta_{\text{b}}^{(\text{p})}(x_{\parallel})$ for $\rho \to \infty$, but as it approaches $\rho \approx 2$ the behaviour changes dramatically; the minimum shifts towards the unordered phase, and grows again between $\rho \approx 2$ and $\rho \approx 1$ to converge towards the scaling function

$$\Theta_{\perp}^{(++)}(x_{\perp}) = \Theta_{\perp}^{(\text{oo})}(-x_{\perp}) \tag{5.21}$$

of the (++) film, see Figs. 16 and 17. As this limit is already approached at $\rho \approx 1$, consequently the systems behaviour does not change for $0 \leq \rho \leq 1$, as shown in Fig. 18, where only $\Theta_{\perp}^{(++,\text{p})}(x_{\perp})$ (for $\rho = 1$ and $\rho = 1/2$) and the thin film limit $\Theta_{\perp}^{(++)}(x_{\perp})$ are shown. Because of the polarisation effect of the surfaces, the scaling functions go to zero for $|x_{\parallel}| \to \infty$, as the $Z_2$-symmetry of the system is broken and no spontaneous flip of the magnetisation is possible.

For the (+−) BCs the behaviour is very similar to any other system that forms a domain wall along one direction, the only difference here being that it is imposed by boundary fields and not an antiperiodic boundary. Nevertheless there is a linear divergence in the ordered phase due to the aforementioned domain wall, but additionally the surface fields give a logarithmic correction. Just as for the (++) boundaries, the scaling function approaches its thin film limit $\Theta_{\perp}^{(+-)}$ almost already for $\rho \approx 1$, see Fig. 20, while in the contrary limit the system is dominated by $\Theta_{\text{b}}^{(\text{p})}(x_{\parallel})$ again, that is the periodicity in the parallel direction as shown in Fig. 19.

## 6 Staggered BCs on both sides

Finally we impose staggered boundary fields to both the upper and lower boundary of the system. In the scaling limit, this setup is believed to be equivalent to Dirichlet and open boundary conditions (oo) [33]. We implement these BCs by an infinitely strong negative coupling in the corresponding additional line and by a construction analogous to the (++) and (+−) BCs. The negative coupling within this additional line is equivalent to exchanging

$$\mu_0^{\pm}(\varphi) \to \mu_0^{\mp}(\varphi), \tag{6.1}$$

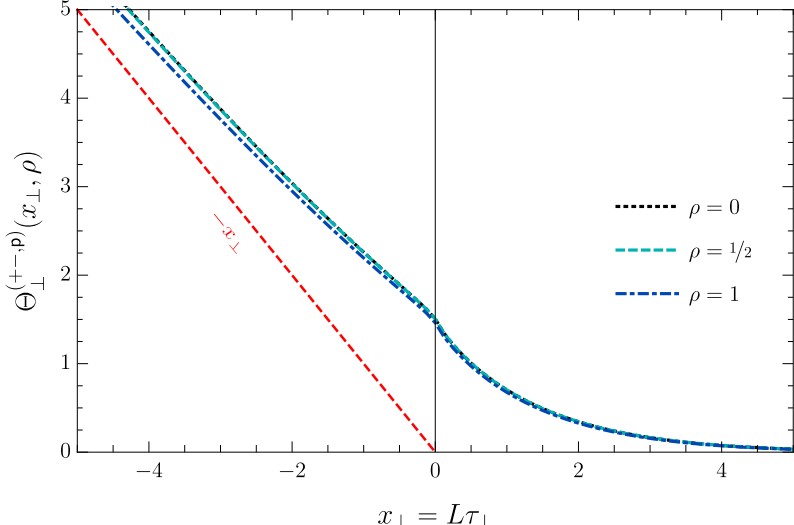

Figure 20: Scaling function $\Theta_\perp^{(+-,\mathrm{p})}(x_\perp,\rho)$ for different values of the aspect ratio $\rho \leq 1$. The scaling function converges rather fast towards its limiting case, which is a linear divergence in the ordered phase plus logarithmic corrections.

which leads to the matrix

$$
\mathcal{C}_{L+1}^{(\uparrow\downarrow\uparrow\downarrow)}(\varphi_m) = 
\begin{pmatrix}
a_1 & -1 & & & b_{L+1} \\
-1 & & & & \\
& & \mathcal{C}_{L-1}^{(0)} & & \\
& & & & b_L \\
b_{L+1} & & & b_L & a_{L+1}
\end{pmatrix},
\tag{6.2a}
$$

with the boundary entries

$$
a_1 = \frac{t_-}{z}\left[\mu^+(\varphi_m) - z_l^2\mu_0^+(\varphi_m)\right], \qquad\qquad b_{L+1} = 0, \tag{6.2b}
$$

$$
a_{L+1} = \frac{t_-}{z}\left[\mu_0^-(\varphi_m) - z_r^2\mu^-(\varphi_m)\right], \qquad\qquad b_L = -\frac{z_r}{z}. \tag{6.2c}
$$

The determinant can thus be calculated with the method of Appendix B and reads

$$
\det\mathcal{C}_{L+1}^{(\uparrow\downarrow\uparrow\downarrow)}(z_l,z_r;\varphi_m) = \frac{t_-}{z}\mu_0^-(\varphi_m)\,\mathrm{e}^{L\gamma_m}\frac{\eta_+^{(\mathrm{oo})}(\varphi_m)\,\eta_+^{(\uparrow\downarrow)}(z_l,\varphi_m)\,\eta_+^{(\uparrow\downarrow)}(z_r,\varphi_m)}{2\sinh\gamma_m} \times
$$

$$
\times\left[1 + \frac{\eta_-^{(\mathrm{oo})}(\varphi_m)\,\eta_-^{(\uparrow\downarrow)}(z_l,\varphi_m)\,\eta_-^{(\uparrow\downarrow)}(z_r,\varphi_m)}{\eta_+^{(\mathrm{oo})}(\varphi_m)\,\eta_+^{(\uparrow\downarrow)}(z_l,\varphi_m)\,\eta_+^{(\uparrow\downarrow)}(z_r,\varphi_m)}\,\mathrm{e}^{-2L\gamma_m}\right].
\tag{6.3}
$$

As due to the infinitely-strong coupled line of spins the matrices for periodic and antiperiodic BCs in perpendicular direction do not differ, only two determinants are left for the non-regular part of the partition function. Thus the form is analog to the $(++)$ and $(+-)$ BCs and reads

$$
\frac{Z^{(\uparrow\downarrow\uparrow\downarrow,\mathrm{p})}}{Z_0^{(\uparrow\downarrow\uparrow\downarrow,\mathrm{p})}} = \frac{1}{2}\left(\frac{2z}{1+t_+}\right)^{\frac{LM}{2}}\left(-\frac{2z}{t_-}\right)^{\frac{M}{2}} \times
$$

$$
\times\left[\prod_{\substack{0<m<2M \\ m\,\mathrm{odd}}}\det^{\frac{1}{2}}\mathcal{C}_{L+1}^{(\uparrow\downarrow\uparrow\downarrow)}(\varphi_m) + \prod_{\substack{0\leq m<2M \\ m\,\mathrm{even}}}\det^{\frac{1}{2}}\mathcal{C}_{L+1}^{(\uparrow\downarrow\uparrow\downarrow)}(\varphi_m)\right],
\tag{6.4}
$$

while the case of opposite row-wise surface fields ($\uparrow\downarrow\downarrow\uparrow$) is given as difference of the products.

In order to demonstrate the equivalence of the doubly staggered ($\uparrow\downarrow\uparrow\downarrow$) and the open (oo) BCs in the scaling limit, we proceed as we did for the symmetry-breaking BCs in the last section. First we factorise the odd contribution which is, analogously to the system with only one staggered surface field, equal to the scaling function of the open cylinder. Thus we have to show that the remaining terms vanish in the scaling limit, as the alternating contribution is exponentially suppressed with growing system size. We do so by a closer look at the bulk contribution, which reads

$$L \ln t + \sum_{m=1}^{M-1} (-1)^m \gamma_m = \frac{L}{2} \sum_{m=1}^{2M} (-1)^m \gamma_m + L \ln[\min(t, z)\, t]. \tag{6.5}$$

The terms that differ in the ordered and the unordered phase have the same scaling form,

$$L \ln(zt) \simeq L \ln(t^2) \simeq -2L \ln a_\xi, \tag{6.6}$$

due to the continuity of the free energy at this point. Nevertheless, the leading contribution diverges linear in $L$, which gives an exponential decay in the related contribution to the free energy scaling function,

$$\ln\left[ 1 \pm e^{L \ln t + \ln\left(2t_- z_{\mathrm{st}}^2 / z^2\right) + \sum_{m=1}^{M-1} (-1)^m \det \mathcal{C}_{L+1}^{\uparrow\downarrow\uparrow\downarrow}} \right] \to 0, \tag{6.7}$$

and thus the free energy of a system with doubly staggered BCs ($\uparrow\downarrow\uparrow\downarrow$)/($\uparrow\downarrow\downarrow\uparrow$) converges either from above or below against the scaling function for the open cylinder, depending on whether the staggered fields are shifted against each other or not, while the corresponding scaling functions in direction $\delta \in \{\perp, \parallel\}$ are equal to the Dirichlet case,

$$\Theta_\delta^{(\uparrow\downarrow\uparrow\downarrow,\mathrm{p})}(x_\delta, \rho) = \Theta_\delta^{(\uparrow\downarrow\downarrow\uparrow,\mathrm{p})}(x_\delta, \rho) = \Theta_\delta^{(\mathrm{oo},\mathrm{p})}(x_\delta, \rho). \tag{6.8}$$

# 7 The antiferromagnetic Ising model

While all results in this work as well as in [1] are obtained for the ferromagnetic (FM) anisotropic Ising model, $K^\delta > 0$, they can easily be generalised to antiferromagnetic (AFM) couplings by the two transformations

$$\{K^\perp, \sigma_{\ell,m}\} \mapsto \{-K^\perp, (-1)^\ell \sigma_{\ell,m}\}, \tag{7.1a}$$

$$\{K^\parallel, \sigma_{\ell,m}\} \mapsto \{-K^\parallel, (-1)^m \sigma_{\ell,m}\}, \tag{7.1b}$$

which, for both directions, reverse the couplings and flip every second spin, while leaving the Hamiltonian invariant. Note that we get a striped AFM system if only one of the two transformations is applied. At the boundaries, these transformation have to be applied, too, where they map the BCs according to the following:

**AFM bonds $K^\perp < 0$:** While for even $L$ all BCs are unchanged under mapping (7.1a), for odd $L$ only the left BC $\alpha_l$ is unchanged and the right BC changes sign, $(\alpha_l +) \leftrightarrow (\alpha_l -)$ and $(\alpha_l \uparrow\downarrow) \leftrightarrow (\alpha_l \downarrow\uparrow)$. Only the open BC is invariant under (7.1a), $(\alpha_l \mathrm{o}) \mapsto (\alpha_l \mathrm{o})$.

**AFM bonds $K^\parallel < 0$:** The mapping (7.1b) is a little bit more complicated, as it can change the BCs in both directions: for even $M$ the BC $\beta$ in parallel direction is unchanged, while the BCs in perpendicular direction are modified independently on both sides according to $(+) \leftrightarrow (\uparrow\downarrow)$, $(-) \leftrightarrow (\downarrow\uparrow)$. Only the open case is invariant, $(\mathrm{o}) \mapsto (\mathrm{o})$. If $M$ is odd, in addition to these rules the BC in parallel direction is changed from periodic to antiperiodic and vice versa, $(\mathrm{p}) \leftrightarrow (\mathrm{a})$.

Both rulesets can be applied independently, such that for the fully AFM Ising model with $K^{\parallel,\perp} < 0$ the combination of both transformations (7.1) yields, e. g., the mapping $(++,\mathsf{p}) \mapsto (\uparrow\downarrow\uparrow, \mathsf{a})$ if both $M$ and $L$ are odd.

We emphasise that (7.1b) changes symmetry breaking BCs like $(++)$ into symmetry preserving BCs $(\uparrow\downarrow\uparrow\downarrow)$ and vice versa, such that for the AFM Ising model the staggered BCs become symmetry breaking. Furthermore, in this model we predict a strong dependency on the shift in the doubly staggered BCs, as $(\uparrow\downarrow\uparrow\downarrow)$ and $(\uparrow\downarrow\downarrow\uparrow)$ in the AFM model correspond to $(++)$ and $(+-)$ BCs in the FM model, leading to negative or positive Casimir potentials and attractive or repulsive Casimir forces, see also [54].

# 8 Conclusions

We presented a systematic calculation of the universal free energy scaling functions for various anisotropic $2d$ Ising systems. Therefore we started with the scaling functions for the periodic, open cylinder. Then we turned to the antiperiodic cylinder and calculated the surface tension of the induced domain wall within our formalism. Afterwards we showed how a boundary field can be introduced within the dimer approach and calculated the corresponding contributions in both the thermodynamic limit as well as the scaling limit for homogenous and staggered surface fields. Using the results from the previous calculations for the homogenous boundary field together with the schemes for periodic and antiperiodic boundary conditions on the torus, we introduced the aspect-ratio-depended form of the free energy scaling function for $(++)$ and $(+-)$ boundary conditions. Finally we translated those results to the case of staggered surface fields, and showed that their scaling limit is indeed equivalent to the one for open boundaries.

During these calculations we saw how the free energy of the two-dimensional Ising model decays into different blocks, and that the scaling functions decay further than previously assumed. This behaviour should indeed translate to more complex systems, e. g., higher spatial dimensions, more complex geometric constrains, or other universality classes. Especially the question if the conformal invariance at criticality can be expanded towards or at least used as approximation of the scaling limit can now be tangled by a comparison with colloidal suspensions.

# Acknowledgements

We like to thank Malte Henkel for helpful discussions and inspirations. We thank the anonymous referees for many helpful suggestions.

**Funding information**  This work was supported by the Deutsche Forschungsgemeinschaft through Grant HU 2303/1-1.

# A Series expansions

In this appendix we show the details of the series expansion necessary to calculate the scaling forms of the contributions to the open boundary surface, the homogeneous, and the staggered boundary field. The expansions are done with the substitution $M = \epsilon^{-1}$ around $\epsilon = 0$, giving the limit for large $M$. We start with the Onsager dispersion (I.4.2), which we already expanded

in (I.4.18) in $\epsilon$ up to the second order as

$$\cosh\gamma = 1 + \frac{y^2}{2}\epsilon^2 + \mathcal{O}(\epsilon^4), \tag{A.1a}$$

with the abbreviation

$$y = \sqrt{\frac{\Gamma^2 + \epsilon x_\parallel \Phi^2}{r_\xi^2 + \epsilon r_\xi x_\parallel}}. \tag{A.1b}$$

**Open boundaries**

The contribution for open boundaries consist of the three parts $e^{\pm\gamma}$, $\sinh\gamma$, and $t_-\mu^+(\varphi)/z$, which can be expanded independently. The first one thus reads

$$e^\gamma \approx \left(1 + \epsilon^2 \frac{y^2}{2}\right) + \left(\epsilon^2 \frac{y^2}{2}\right)^{\frac{1}{2}} \left(2 + \epsilon^2 \frac{y^2}{2}\right)^{\frac{1}{2}} \tag{A.2a}$$

$$= 1 + \epsilon\, y \sqrt{1 + \epsilon^2 \frac{y^2}{4}} + \epsilon^2 \frac{y^2}{2} \tag{A.2b}$$

$$= 1 + \epsilon\, \frac{\Gamma}{r_\xi} + \mathcal{O}(\epsilon^2), \tag{A.2c}$$

where we used the identity

$$e^{\operatorname{arcosh} x} = x + (1-x)^{\frac{1}{2}} (1+x)^{\frac{1}{2}}. \tag{A.3}$$

Consequently, the inverse can be expanded as

$$e^{-\gamma} = 1 - \epsilon\, \frac{\Gamma}{r_\xi} + \mathcal{O}(\epsilon^2). \tag{A.4}$$

The second term can be simplified by the identity

$$\sinh\operatorname{arcosh} x = (1+x)(1-x)^{\frac{1}{2}} (1+x)^{-\frac{1}{2}} \tag{A.5}$$

as

$$\sinh\gamma \approx \left(2 + \epsilon^2 \frac{y^2}{2}\right)\left(\epsilon^2 \frac{y^2}{2}\right)^{\frac{1}{2}} \left(2 + \epsilon^2 \frac{y^2}{2}\right)^{-\frac{1}{2}} \tag{A.6a}$$

$$= \epsilon\, y \left(1 + \epsilon^2 \frac{y^2}{4}\right)^{\frac{3}{2}} \tag{A.6b}$$

$$= \epsilon\, \frac{\Gamma}{r_\xi} + \mathcal{O}(\epsilon^2). \tag{A.6c}$$

Finally the last term reads

$$\frac{t_-}{z}\mu^+(\varphi) = \frac{t_+}{z} - \frac{t_-}{z} + \epsilon^2 \frac{t_-}{z}\Phi^2 + \mathcal{O}(\epsilon^4) \tag{A.7a}$$

$$= \frac{t}{z} + \mathcal{O}(\epsilon^2) \tag{A.7b}$$

$$= 1 + \epsilon\, \frac{x_\parallel}{r_\xi} + \mathcal{O}(\epsilon^2), \tag{A.7c}$$

which can be combined to

$$\ln \frac{\eta_+^{(\text{oo})}(\varphi)}{2\sinh\gamma} \simeq \ln \frac{\Gamma + x_\parallel}{2\Gamma}. \tag{A.8}$$

The corresponding term in the residual strip free energy thus becomes

$$\frac{\eta_-^{(\text{oo})}(\varphi)}{\eta_+^{(\text{oo})}(\varphi)} \simeq \frac{\Gamma - x_\parallel}{\Gamma + x_\parallel} \tag{A.9}$$

in the scaling limit.

**Homogeneous boundary fields**

For the homogeneous BF the only new term is

$$\frac{t_-}{z}\mu_0^-(\varphi) = \frac{t_-}{z}\left[2 - \frac{\epsilon^2}{2}\Phi^2 + \mathcal{O}(\epsilon^4)\right] \tag{A.10a}$$

$$= 2\left[\sqrt{1 + r_\xi^2} - 1\right]^{-1} + \mathcal{O}(\epsilon), \tag{A.10b}$$

with which the associated BF contribution can be expanded to

$$\eta_\pm^{(\text{h})}(z_\text{h}{=}1; \varphi) \simeq 1 \pm \frac{2a_\xi}{\epsilon(\Gamma \pm x_\parallel)} \tag{A.11a}$$

$$= \pm \frac{2a_\xi}{\epsilon(\Gamma \pm x_\parallel)}\left[1 \pm \epsilon\frac{\Gamma \pm x_\parallel}{2a_\xi}\right], \tag{A.11b}$$

with the inverse critical coupling $a_\xi$ from (3.15). We now face the problem that $\eta_\pm^{(\text{h})}(z_\text{h}{=}1; \varphi)$ diverge with growing $M$. But as shown in section 3 the emerging divergences cancel out with the corresponding critical values, thus we have

$$\ln \frac{\eta_\pm^{(\text{h})}(z_\text{h}{=}1; \varphi)}{\eta_\pm^{(\text{h})}(z_\text{h}{=}1; \varphi)\big|_{x_\parallel=0}} \simeq -\ln\frac{\Gamma \pm x_\parallel}{|\Phi|}, \tag{A.12}$$

with the series expansion

$$\ln\eta_+^{(\text{h})}(z_\text{h}{=}1; \varphi)\big|_{x_\parallel=0} \simeq \ln\left(2a_\xi M\right) - \frac{1}{2}\ln(\Phi^2). \tag{A.13}$$

For the residual strip contribution we thus conclude with

$$\frac{\eta_-^{(\text{h})}(z_\text{h}{=}1; \varphi)}{\eta_+^{(\text{h})}(z_\text{h}{=}1; \varphi)} \simeq \frac{\Gamma + x_\parallel}{\Gamma - x_\parallel}. \tag{A.14}$$

# B   Determinant simplifications

Some calculations necessary for the determinant evaluation are neither especially complicated nor particularly enlightening, but nevertheless confusing and deserve a detailed presentation. These calculations are shown in this appendix. Our starting point is the determinant of (3.8a),

$$\det\mathcal{C}_{L+1}^{(\alpha_l \alpha_r)} = a_{L+1}\left[a_1 \det\mathcal{C}_{L-1}^{(0)} - \det\mathcal{C}_{L-2}^{(0)}\right] \\ - b_L^2\left[a_1 \det\mathcal{C}_{L-2}^{(0)} - \det\mathcal{C}_{L-3}^{(0)}\right], \tag{B.1}$$

of a system with BC $\alpha_l/\alpha_r$ on the left/right side, respectively, where we have dropped the dependency on $\varphi_m^{(\beta)}$ to improve readability. Since we need to calculate (B.1) for several different boundary conditions, we notice that the boundary expressions always has the same form,

$$a_1 = \frac{t_-}{z}\left(\mu^+ - A_l\right),\tag{B.2a}$$

$$a_{L+1} = \frac{t_-}{z}\left(A_r - z_r^2\mu^-\right),\tag{B.2b}$$

$$b_L = -\frac{z_r}{z},\tag{B.2c}$$

where $A_l$ and $A_r$ are the terms containing all dependencies for the left and the right boundary field, respectively. Thus we write

$$\det \mathcal{C}_{L+1}^{(\alpha_l\alpha_r)} = \frac{t_-^2}{z^2}\left(A_r - z_r^2\mu^-\right)\left(\mu^+ - A_l\right)\det\mathcal{C}_{L-1}^{(0)} + \frac{z_r^2}{z^2}\det\mathcal{C}_{L-3}^{(0)}$$
$$+ \frac{t_-}{z}\left(\frac{z_r^2}{z^2}A_l - A_r + z_r^2\mu^- - \frac{z_r^2}{z^2}\mu^+\right)\det\mathcal{C}_{L-2}^{(0)},\tag{B.3a}$$

where we can use (I.4.2), namely $\frac{t_-}{z}\left(\mu^+ - z^2\mu^-\right) = 2\cosh\gamma$, as well as the recursion formula $\det\mathcal{C}_{L-1}^{(0)} = 2\cosh\gamma\,\det\mathcal{C}_{L-2}^{(0)} - \det\mathcal{C}_{L-3}^{(0)}$ for matrix (2.5) to further simplify to

$$= \left[\frac{t_-^2}{z^2}\left(A_lz_r^2\mu^- + A_r\mu^+ - A_lA_r\right) - \frac{z_r^2}{z^2}\left(t_-^2\mu^+\mu^- + 1\right)\right]\det\mathcal{C}_{L-1}^{(0)}$$
$$+ \frac{t_-}{z}\left(\frac{z_r^2}{z^2}A_l - A_r\right)\det\mathcal{C}_{L-2}^{(0)}.\tag{B.3b}$$

Now we can use $t_-^2\mu^+\mu^- + 1 = t_-^2\mu_0^+\mu_0^-$ to obtain

$$= \frac{t_-^2}{z^2}\left(A_lz_r^2\mu^- + A_r\mu^+ - A_lA_r - z_r^2\mu_0^+\mu_0^-\right)\det\mathcal{C}_{L-1}^{(0)}$$
$$+ \frac{t_-}{z}\left(\frac{z_r^2}{z^2}A_l - A_r\right)\det\mathcal{C}_{L-2}^{(0)},\tag{B.3c}$$

where we factorise $-t_-/z$ and introduce the abbreviation

$$\eta_{\pm}^{(\alpha)}(z_l, z_r) = \pm\left[\left(\frac{z_r^2}{z^2}A_l - A_r\right)e^{\mp\gamma} + \frac{t_-}{z}\left(A_lz_r^2\mu^- + A_r\mu^+ - A_lA_r - z_r^2\mu_0^+\mu_0^-\right)\right]\tag{B.4a}$$

$$= \pm A_r\left(\frac{t_-}{z}\mu^+ - e^{\mp\gamma}\right) \pm A_l\frac{z_r^2}{z^2}\left(zt_-\mu^- + e^{\mp\gamma}\right)$$
$$\mp \frac{t_-}{z}\left(A_lA_r + z_r^2\mu_0^+\mu_0^-\right)\tag{B.4b}$$

$$= A_r\eta_{\pm}^{(\mathrm{oo})} - A_l\frac{z_r^2}{z^2}\eta_{\mp}^{(\mathrm{oo})} \mp \frac{t_-}{z}\left(A_lA_r + z_r^2\mu_0^+\mu_0^-\right),\tag{B.4c}$$

where we have used $\eta_{\pm}^{(\mathrm{oo})}$ from (2.15).

Finally we find the following results: for the cylinder with one homogeneous BF on the right surface, where $A_l = 0$ and $A_r = \mu_0^+$,

$$\eta_{\pm}^{(\mathrm{o+})}(z_r) = \mu_0^+\,\eta_{\pm}^{(\mathrm{oo})}\,\eta_{\pm}^{(\mathrm{h})}(z_r),\tag{B.5a}$$

for the case of homogeneous BFs on both sides, where $A_l = z_l^2 \mu_0^-$ and $A_r = \mu_0^+$,

$$\eta_\pm^{(\text{hh})}(z_l, z_r) = \mu_0^+ \, \eta_\pm^{(\text{oo})} \, \eta_\pm^{(\text{h})}(z_l) \, \eta_\pm^{(\text{h})}(z_r), \tag{B.5b}$$

and for the case of doubly staggered BCs, where $A_l = z_l^2 \mu_0^+$ and $A_r = \mu_0^-$,

$$\eta_\pm^{(\Uparrow\Downarrow\Uparrow\Downarrow)}(z_l, z_r) = \mu_0^- \, \eta_\pm^{(\text{oo})} \, \eta_\pm^{(\Uparrow\Downarrow)}(z_l) \, \eta_\pm^{(\Uparrow\Downarrow)}(z_r). \tag{B.5c}$$

The resulting determinants then read

$$\det \mathcal{C}_{L+1}^{(\alpha)}(\varphi_m^{(\beta)}) = \frac{t_-}{z} \frac{\eta_+^{(\alpha)}(\varphi_m^{(\beta)}) e^{L\gamma_m^{(\beta)}} + \eta_-^{(\alpha)}(\varphi_m^{(\beta)}) e^{-L\gamma_m^{(\beta)}}}{2\sinh \gamma_m^{(\beta)}} \tag{B.6a}$$

$$= \frac{t_-}{z} e^{L\gamma_m^{(\beta)}} \frac{\eta_+^{(\alpha)}(\varphi_m^{(\beta)})}{2\sinh \gamma_m^{(\beta)}} \left[ 1 + \frac{\eta_-^{(\alpha)}(\varphi_m^{(\beta)})}{\eta_+^{(\alpha)}(\varphi_m^{(\beta)})} e^{-2L\gamma_m^{(\beta)}} \right]. \tag{B.6b}$$

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
