# Peer review of "Anisotropic scaling of the two-dimensional Ising model II: surfaces and boundary fields"

_SciPost Physics, doi:SciPost Phys. 8, 032 (2020)_

## Round 2 · Referee Report · Anonymous (Referee 4) · 2019-7-26

Strengths

Paper gives several new results, not given before.

Weaknesses

There are a few major issues and many minor issues that must be fixed.

Report

This work should be published after corrections. It is hard to read in part due to insufficient detail in citing earlier work. The title of section 6 is wrong: Brascamp-Kunz is periodic in one direction and open in the other with plus boundary one side and plus-minus alternating the other side. This is treated in section 5.

Requested changes

This work has several new results and should be published after some improvements.

Three general comments:

This paper is very hard to read, even for experts, as the authors use their private set of notations defined in earlier papers, leaving the reader to search for needed explanations. Specific and precise citations would be highly helpful.

Citations to just, say [48] (McCoy-Wu book) is not very helpful, as the reader has to search through over 400 pages. Also giving equation numbers as (4.35) is not helping, as there are many such-numbered equations. Give either page number or chapter also, eg. (VI.4.35).

Secondly, throughout the paper the word "according" is used, where one would expect a variety of other words, including "corresponding", "associated", "related", ... .

Thirdly, several times "converge against" is used, which is a "Germanism" (literal translation from German) slipping into English. Proper usage is "converge to".

I will not mention all these occurrences of these three issues.

Specific comments:

Page 2, line 7: "on" should be "one" in "at least one anti periodic".

Page 2, paragraph 3, about middle: "neat" in "Beneath its neat temperature" makes no sense to me. This must be clarified.

Page 3, top: Schur reduction has been used about 50 years ago in similar context by McCoy and Wu, for example in the random-layer Ising model. A suitable comment and/or citation seems appropriate.

Page 3, line 9: "on" should be "one" in "at least one translationally".

Page 3, line 10: it seems better to replace "during" by "in".

Page 3, middle (line 3 of paragraph 2): Replace "arise" (verb) by "emergence" or "appearance" (noun).

Page 4, (2.5): It helps to stress that a factor $-b_1$ is taken out of each element as in the opening of section 4 in paper I.

Page 5, (2.9): It follows directly as (2.4) applied to (2.5) is the recurrence relation of Chebyshev polynomials of the second kind.

Page 5, (2.10) (and possibly somewhere on page 4): It is good to say that in this section nontrivial boundary condition is considered on one side only, and that more general boundary conditions on both sides are addressed in the next section and appendix B.

Page 5: Also opening Appendix B at this point in the paper may confuse the reader. It would be good to add more explanation on the use of subscripts r and l. For example, is $z_r=z_L$ in (B.2), and all other $z_l=z$? Otherwise the reader may at this point search the paper and some cited papers for definitions of $z_r$, etc., wasting a lot of time. The use of r and l seems to appear in the main text first in section 3 (page 17).

Page 5, (2.12): Some added words on $(\alpha,\beta)$ are desired, especially $\beta=p,a$ for the parallel direction and its relation with even/odd in (2.12).

Page 7, middle: Do the authors mean (B.3) in [47]; and to which equation in chapter VI of [48] do they refer?

Page 7, line above (2.25): Replace "find" by "replace the sum over $m$ by a contour integral as in (I.4.27) to find"

Page 7, line below (2.25): "integrand" should be "complex phase of the integrand together with a few contours of equal absolute value"? Those latter values are not given.

Page 13, line below (2.44): Replace $\partial_x$ by $d/dx$ may be clearer, as $\partial_x$ means something special to some readers.

Page 18, second paragraph: Make McCoy-Wu reference more specific. (Au-Yang in J. Math. Phys. 14, 937-946 (1973) also varies the spin-spin coupling in the boundary layer.)

Page 21, figure caption end: "an" must be "and".

Page 21, two lines above (3.16): "therefor" should be "therefore".

Page 22, (3.19): Better replace $\partial_s$ by $d/ds$.

Page 25, above (4.5): Term "macro spin" may be confusing for staggered case?

Page 29, (5.14): Better replace $\partial_s$ by $d/ds$.

Page 29, bottom lines: "contribution to" and twice "sum for". The sums are over $m$.

Page 31, line below (5.20): "obtained".

Page 31, 3 lines lower: Delete "the" before "several".

Page 31, line above (5.21): Replace "against" by "toward".

Page 32, line below figure: "fields give" or "field gives".

Page 32, Section 6: This is not the conditions in the paper of Brascamp and Kunz. They had one side all plus and the other side alternating plus-minus, in which case the temperature zeros of the partition function are particularly simple. This corresponds to (5.3). Please check Figure 1 in J. Math. Phys. 15, 65-66 (1974). The authors must correct this.

Page 33, caption figure 20: "agains" should be "to" or "again to".

Page 34, line 2 section 7: "Therefor" -> "Therefore"

Page 34, 4 lines from bottom: "spatial" is preferred spelling.

---

## Round 2 · Referee Report · Anonymous (Referee 3) · 2019-9-6

Strengths

1-This paper reports new results on the scaling functions at finite aspect ratio of the two-dimensional Ising model with various boundary conditions.

Weaknesses

1-There are some lacks of clarity, issues with referencing, and multiple issues having to do with the English.

Report

Building on a previous paper by the same authors, new expressions for the scaling functions of the two-dimensional Ising model with various boundary conditions are given. Notably these results are valid at arbitrary temperatures and aspect ratios, and so complement existing CFT results as well as lattice-model analyses at non-finite aspect ratio. Although the analytical techniques used are of a well-known type (close to the work by McCoy and Wu notably) the results are interesting to experts in this field.

The paper could however gain significantly in clarity by clearing a few issues.

First of all, the introduction should state more clearly which principal results are claimed as new. A greater effort should be made throughout in referring to the existing literature in a precise way. To give but one example, the reference to McCoy and Wu (ref. [48]) after eq. (2.22) is close to useless if the authors do not state precisely where in this comprehensive monograph to look. Some readers might also dislike the authors' use of their own previous article, from which they simply use several numbered equations without making the effort of briefly reviewing how they were obtained or why they should be used in the present context.

Second, the authors maul the English language in several ways that decidedly diminish the reader's comprehension and obscure some passages. While it would be too time consuming to give a full list, here are a few examples. The word "according" is used in many places where another word would be clearer and more appropriate; there is a confusion between "on" and "one"; expressions like "remindful of" and "frankly expand" make no sense. Furthermore, the use of more specialised terminology is often questionable. For example, is a "transition circle" what Kasteleyn originally called a "transition cycle"? Is a "keyhole integral" some well-established terminology (and where), or does it vaguely refer to an integral along a contour that has the shape of a keyhole (in which case it is certainly not very precise)? Note also that the use of "Brascamp-Kunz boundary conditions" in section 6 and elsewhere is not identical to the precise boundary conditions coined by these authors; maybe it would be better simply to say "staggered boundary conditions".

In line 2 of section 4, the authors claim that staggered boundary conditions are believed to be equivalent to Dirichlet, but they fail to give a precise reference.

To increase the interest of the paper, I would suggest that the authors also analyse their scaling functions in the vicinity of the antiferromagnetic transition of the Ising model. There has been some recent results in the context from the CFT perspective, see arXiv:1906.07565 by Robertson et al. In particular, when staggered boundary conditions are imposed on both sides of the strip, it turned out to matter whether they are correlated (+-+-... vs +-+-...) or anti-correlated (+-+-... vs -+-+...), and it would be interesting to see if the authors find similar results.

Requested changes

1-Improve referencing
2-Solve English and terminology issues
3-Clarify the introduction
4-Comment on the antiferromagnetic transition

---

## Round 3 · Referee Report · Anonymous (Referee 1) · 2019-12-29

Strengths

Spelled out in previous report.

Weaknesses

I believe they have been addressed since the previous report. Still not easy to read, but that is to be expected, as the calculations are involved.

Report

Publish this version, unless the other referee sees something still.

Requested changes

No changes suggested

---

## Round 3 · Referee Report · Anonymous (Referee 2) · 2020-1-14

Strengths

As in my previous report.

Weaknesses

The authors have improved the clarity of the paper by implementing the suggestions of the two referees.

Report

I recommend acceptance in the present form. (The authors may however wish to mend a tiny BibTeX related issue regarding the title of reference [38].)

---

## Round 3 · Author Response

Dear editors,

we thank both referees for their comprehensive reports and changed the manuscript accordingly. Both reports were really helpful and increased the readability of the manuscript. We made all requested changes as recommended, see below, with only a two exceptions: Schur reductions are common in mathematical physics, hence we did not cite specific papers, and the term "keyhole integral" is also common in complex integration where cuts are involved, so we did not change this.
Furthermore, we rewrote several sentences in order to increase the readability of the manuscript. A list of changes is given below.

---

## Round 3 · List of Changes

Changes as requested in Report 1:

o) changed the incorrectly used wordings as "according", "converge against", and so on, and fixed many typos

o) added a remark to Chebyshev polynomials near (2.9)

o) better explained the use of subscript "l" and "r"

o) better explained α and β

o) fixed Refs. [47,48]

o) fixed the formulation above (2.25)

o) fixed the contour lines in the complex contour plots Figs. 2, 5, 11, 15. We are grateful to referee 1 for pointing out the ambiguous use of contour lines. Now "the lines of constant absolute value c are shown as black dotted (c < 1), dashed (c = 1) or solid (c > 1) lines, where c are consecutive integer powers of two."

o) replaced ∂x by d/dx throughout the manuscript

o) fixed the wording concerning the sum over m on page 29

o) renamed the staggered boundary condition from "Brascamp-Kunz" to "staggered" throughout the paper. We agree with both referees that "staggered" and "doubly staggered" better reflects these boundary conditions.

Changes as requested in Report 2 (if not already noted above):

o) fixed the spelling error "transition circle" to "transition cycle". Thank you for this hint.

o) added references regarding the equivalence of Dirichlet, staggered and open boundary conditions

o) Finally, we thank referee 2 for the reference to arXiv:1906.07565. We added section 7 "The antiferromagnetic Ising model" and discussed the transformation from our results to the antiferromagnetic Ising model.

---

## Editorial Decision

published